# Evaluating Relational Reasoning in LLMs with REL

**Lukas Fesser** [* 1]  **Yasha Ektefaie** [* 2]  **Ada Fang** [* 1]  **Sham Kakade** [1]  **Marinka Zitnik** [1]

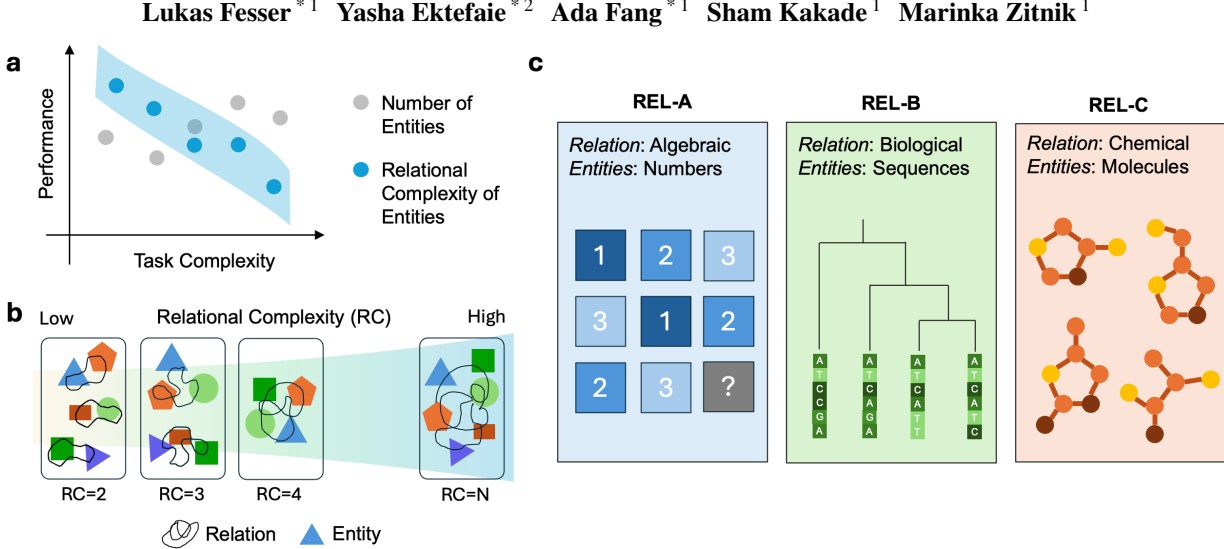

*Figure 1:* **a** Performance decreases as relational complexity increases, even when the number of entities varies across tasks. Entity count is therefore a noisy proxy for task difficulty. **b** Relational complexity increases with the number of entities that must be jointly bound to satisfy a shared constraint, i.e., when correctness depends on a higher-arity relation. **c** **REL** evaluates relational reasoning in LLMs across algebra, biology, and chemistry.

## Abstract

Relational reasoning is the ability to infer relations that jointly bind multiple entities, attributes, or variables. This ability is central to scientific reasoning, but existing evaluations of relational reasoning in large language models often focus on structured inputs such as tables, graphs, or synthetic tasks, and do not isolate the difficulty introduced by higher-arity relational binding. We study this problem through the lens of *Relational Complexity (RC)*, which we define as the minimum number of independent entities or operands that must be simultaneously bound to apply a relation. RC provides a principled way to vary reasoning difficulty while controlling for confounders such as input size, vocabulary, and representational choices. Building on RC, we introduce **REL**, a generative benchmark framework spanning algebra, chemistry, and biology that varies

RC within each domain. Across frontier LLMs, performance degrades consistently and monotonically as RC increases, even when the total number of entities is held fixed. This failure mode persists with increased test-time compute and in-context learning, suggesting a limitation tied to the arity of the required relational binding rather than to insufficient inference steps or lack of exposure to examples. Our results identify a regime of higher-arity reasoning in which current models struggle, and motivate re-examining benchmarks through the lens of relational complexity.

🌐 Project Page    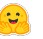 GitHub    🤗 Hugging Face

## 1. Introduction

Relational reasoning, the process of inferring an unknown from the interaction of multiple entities and relations, is widely regarded as a core component of human reasoning (Halford et al., 1998; Alexander et al., 2016; Dumas et al., 2013). Despite recent progress on a broad range of reasoning tasks, Large Language and Reasoning Models (LLMs and LRMs) are rarely evaluated on relational reasoning (Clark et al., 2018; Cobbe et al., 2021; Hendrycks et al., 2021; Geva et al., 2021; Liévin et al., 2024; Wang et al., 2023b). Research that studies relational reasoning in these

---

*Equal contribution  [1]Harvard University [2]Eric and Wendy Schmidt Center at the Broad Institute of MIT and Harvard. Correspondence to: Marinka Zitnik <marinka@hms.harvard.edu>.

*Proceedings of the 43rd International Conference on Machine Learning*, Seoul, South Korea. PMLR 306, 2026. Copyright 2026 by the author(s).

models focuses on graph-based settings, including networks and knowledge graphs (Wang et al., 2023a; Tang et al., 2024; Wu et al., 2025; Zhang et al., 2024; Fatemi et al., 2024), or on multi-hop reasoning over knowledge bases (Yang et al., 2018; Ho et al., 2020; Trivedi et al., 2022). This leaves open a broader question: *how well do state-of-the-art LLMs/LRMs handle relational structure in scientific reasoning*, such as numerical pattern rules, algebraic constraints, and chemical structure regularities? We address this question with a unified set of tasks that varies relational complexity while controlling for other sources of difficulty, including entity complexity, across three scientific domains.

In many scientific problems, the answer is not carried by any single cue. Instead, a model must combine multiple constraints that link variables, measurements, or symbols to reach a correct conclusion (Bemis & Murcko, 1996; Stern, 2013). When we cannot measure where models break under this multi-constraint integration, benchmark performance becomes difficult to interpret: strong results may reflect saturation rather than progress on harder forms of reasoning (Deveci & Ataman, 2025). Identifying unsaturated dimensions of reasoning performance is necessary to understand where current models still fail and where further improvements are possible. Evaluating this capability is challenging because standard proxies for "difficulty" do not isolate the relational bottleneck. Performance can drop for reasons that have little to do with relational reasoning, such as longer prompts, different prompt templates, or additional background knowledge being required. As a result, evaluations often mix the cost of parsing and retrieval with the cost of relational inference. Existing studies have largely evaluated relational reasoning through graph-centric formulations (Liu et al., 2025a). Beyond graph tasks, we still lack a task-agnostic notion of relational difficulty.

Here we introduce *Relational Complexity* (RC) as a principled measure of how many independent operands and entities must be represented and jointly composed to solve a task (Fig. 1b). Importantly, relational complexity has a history outside of machine learning, where it is used in cognitive science and related fields (Halford et al., 1998; Carpenter et al., 1990; Crone et al., 2009) to characterize reasoning demands. In this paper, we present **REL**, a suite of tasks where relational complexity can be controlled. Our approach has the following advantages: **(1) Parameterization of RC for scientific reasoning.** We design tasks across mathematics, biology, and chemistry that allow systematic control over relational complexity (Fig. 1c). **(2) Isolating the effect of RC on performance.** We isolate the impact of RC on LLM performance by controlling for and marginalizing over confounding factors. **(3) A scalable framework for probing scientific relational reasoning.** We introduce a generative benchmark that produces novel questions at systematically increasing levels of difficulty,

enabling evaluation across model capabilities.

Evaluating frontier LLMs on **REL**, we observe that performance degrades sharply as relational complexity increases across mathematical, biological, and chemical domains. Measures of RC are complementary to performance changes observed in size-based proxies including input length and entity count (Fig. 1a). For Raven's matrices and tensors in **REL**-A, performance drops by 45% when RC increases from 3 to 9. Applying RC to reasoning over evolutionary history with **REL**-B1, we find increasing RC from 5 to 20 leads to accuracy decreasing by 93%. In the chemistry tasks, task completion rate decreases by 39.7% from **REL**-C1 to **REL**-C3 when models reason over molecules represented as SMILES. Relational complexity reveals a clear and consequential limitation of current state-of-the-art models.

## 2. Related Work

**Reasoning Benchmarks in Machine Learning.** Reasoning benchmarks in machine learning have expanded rapidly alongside the development of LLMs and LRMs. Existing benchmarks span a wide range of capabilities, including arithmetic and symbolic manipulation (Hendrycks et al., 2021; Cobbe et al., 2021), program synthesis (Austin et al., 2021; Chen, 2021), logical deduction (Clark et al., 2018; Liu et al., 2020; Tafjord et al., 2021), tool use (Qin et al., 2024; Li et al., 2023), and multi-hop question answering (Yang et al., 2018; Ho et al., 2020; Trivedi et al., 2022). A key limitation of existing benchmarks is the lack of explicit control over the relational structure underlying a task. Recent work has begun to probe LLMs' ability to reason over graphs (Wang et al., 2023a; Tang et al., 2024; Wu et al., 2025; Zhang et al., 2024; Fatemi et al., 2024) and knowledge graphs (Edge et al., 2024; Zhu et al., 2025; He et al., 2024; Li et al., 2024a; Luo et al., 2023). Liu et al. (2025a) introduces a generative benchmark for relational reasoning, but their framework focuses on changing the underlying relational graph structure of tasks. While valuable, these settings rely on graph-specific formalisms that do not generalize cleanly to broader scientific reasoning tasks and often vary surface representations without systematically altering the underlying relational complexity.

**Relational Reasoning in Cognitive Science.** In cognitive science, relational reasoning is commonly studied as the ability to represent relations among entities, align roles across multiple structures, and compose several relations to infer a missing element or choose a consistent completion (Alexander et al., 2016; Dumas et al., 2014b; Carpenter et al., 1990; Halford et al., 1998; Crone et al., 2009). Tasks such as Raven's Progressive Matrices and related analogical paradigms are widely used (Brouwers et al., 2009; Mills et al., 1993; Burke, 1972; Williams & McCord, 2006) because they require extracting abstract relational structure

that generalizes beyond surface features, and they permit controlled manipulations of the number of relations that must be simultaneously maintained. A closely related concept is relational complexity (Halford et al., 1998), which characterizes task difficulty by the number of independent "slots" (entities or variables) that must be bound and processed concurrently to perform the required inference. This framing separates relational difficulty from superficial complexity and helps explain sharp changes in performance as tasks move from binary to higher-arity relational integration. In this paper, we leverage it to motivate a model-agnostic difficulty axis for evaluating LLM/LRM relational reasoning. We provide an extended related work in Appendix B.

## 3. Defining Relational Complexity

Here we formalize the concept of Relational Complexity using the example of Raven's Progressive Matrices (RPMs) (John & Raven, 2003), before introducing the individual components of **REL** which spans arithmetic, biology, and chemistry (Fig. 2).

**Definition of Relational Complexity (RC).** Relational complexity (RC) is the minimal number of independent sources of variation that must be bound and represented at the same time to carry out a reasoning step (Fig. 1b). Independent sources are variables that can change freely and must be tracked separately. Binding links specific fillers to distinct argument roles, which are the slots a relation provides. The required representations span as many dimensions as there are such sources, and the relational complexity equals the relation's arity, meaning the number of argument roles that must be handled together.

**Definition of Operand Complexity (OC).** Operand complexity (OC) refers to the difficulty of identifying, representing, or inferring the fillers that occupy argument roles in a relation, independent of the number of roles themselves. Tasks can share the same RC but have different OC.

We use **Raven's Progressive Matrices** (RPMs) (John & Raven, 2003) to illustrate this definition. In their original form, RPMs are tasks that associate vision with relational and analogical reasoning in a hierarchical representation. They come with three different levels of difficulty defined by RC, and were introduced to test cognitive abilities in children and young adults (Carpenter et al., 1990). Consider the three examples of RPMs in Fig. 3, below we describe how we obtain the RC of each matrix.

In $RPM_1$ of Fig. 3, no row or column relation is present, only matching. The solver holds a single template feature (e.g., "solid upright triangle") and scans the three candidates until one is identical. Only one independent source of variation, the template itself, must be represented in parallel to

determine the answer. Hence, the bottleneck is unary, giving RC $= 1$.

In $RPM_2$ of Fig. 3, the rule is "horizontal or vertical," so the blank can be solved by using one axis only, either the row or the column. For any active attribute (here, the semicircle's orientation), the two known cells along the chosen axis determine the third. Thus, the bottleneck is binary, giving RC $= 2$.

In $RPM_3$ of Fig. 3, the missing cell must satisfy both the row rule and the column rule at the same time. For any active attribute (e.g., the symbol's type or marking), the value in the blank is determined by integrating the two known cells in its row with the two known cells in its column. Four independent operands must be held in parallel, so the bottleneck is quaternary, giving RC $= 4$.

## 4. Relational Reasoning Benchmark

### 4.1. Relational Reasoning in Algebra (REL-A)

RPMs do not need to involve visual components, and we can reduce them to symbolic tasks by representing matrices using their attributes, e.g. "a black triangle in column one, row one" as done in (Hersche et al., 2025), or by working directly with numerical arrays as in Fig. 2 (Camposampiero et al., 2025b). Unlike (Camposampiero et al., 2025a;b), we do not use confounders or noise to confuse the model since we are primarily interested in relational reasoning capabilities as measured via relational complexity, which neither confounders nor perceptual noise affect. We note that the missing entry in an RPM is a function of the remaining entries, so we can directly control the relational complexity of the task by increasing the number of entries on which that function's output depends.

To design more difficult RPMs with a relational complexity much greater than 4, we introduce a generalization we call **Raven's Progressive Tensors (RPT)s**. To solve the RPT, models must reason over functions and rules where the missing value depends on the one-hop neighborhood, such as the sum of adjacent values. With higher dimensions, we can achieve $RC_{2-dim} \leq 8$, $RC_{3-dim} \leq 26$, $RC_{4-dim} \leq 80$, $\ldots$, $RC_{n-dim} \leq 3^n - 1$. In our experiments, we use the following seven rules to generate RPMs and RPTs with varying relational complexities.

Suppose we are given a $n \times n$ RPM: **A1 (Constant).** Each entry in the RPM are the same value. RC $= 1$. **A2 (Progression).** Each entry is the value of its predecessor in the same row plus a fixed value. RC $= 2$. **A3 (Permutation).** Each row contains the same $n$ values in a random (non-repeating) order. RC $= n$. **A4 (Row-Sum).** The final value in each row is the sum of all other entries in the same row multiplied by either $\pm 1$, depending on the column. RC $= n$. Now suppose we are given a $n \times n \times n$ RPT: **A5 (4-Moving-**

## Algebra

**REL-A2 (RC=2)**

**Q:** Only return the missing number. [1, 4, 7, 10, 13, 16] | [5, 8, 11,14, 17, 20] | [3, 6, 9, 12, 15, ?]

**Raven Matrix**
Progression

| 1 | 4 | 7 | 10 | 13 | 16 |
|---|---|---|----|----|----|
| 5 | 8 | 11 | 14 | 17 | 20 |
| 3 | 6 | 9 | 12 | 15 | ? |

**A:** 18

**REL-A3 (RC=6)**

**Q:** Only return the missing number. [1, 2, 3, 4, 5, 6] | [5, 3, 2, 6, 4,1] | [4, 3, 6, 2, 1, ?]

**Raven Matrix**
Permutation

| 1 | 2 | 3 | 4 | 5 | 6 |
|---|---|---|---|---|---|
| 5 | 3 | 2 | 6 | 4 | 1 |
| 4 | 3 | 6 | 2 | 1 | ? |

**A:** 5

**REL-A7 (RC=6)**

**Q:** Only return the missing number. depth 1: [3, 2, 5] | [1, 6, 2] | [1,1, 3]; depth 2: [?, 2, 1] | [5, 0, 2] | [6, 5, 0]; depth 3: [4, 6, 6] | [5, 1, 6]| [2, 5, 4]

**Raven Matrix**
Neighborhood Sum

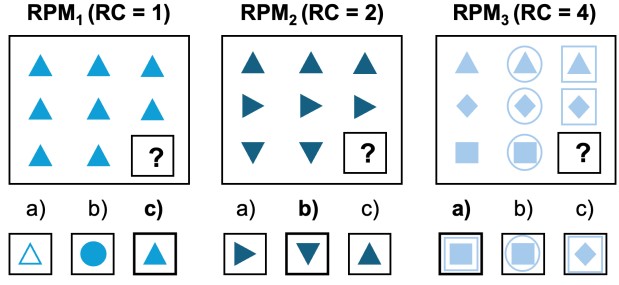

**A:** 0

## Biology

**REL-B1 (RC=2)**

**Q:** Homoplasy refers to structured convergence, pairs or groups of distantly related taxa that repeatedly share the same nucleotide motifs, across many independent alignment columns more often than expected, while other taxa with similar overall sequences do not share those nucleotide motifs as consistenty. Your job is to examine the entire alignment and provided tree and decide whether such structured homoplasy is likely to be present and which taxa are involved. Return your answer as: Yes/No and if Yes, list the taxa involved.

**Phylogenetic Tree (Represented in Newick tree format)**

((((A:0.2,((B:0.1,C:0.1):0.1,D:0.2):0.1):0.2, (E:0.2,(F:0.1,G:0.1):0.1):0.2):0.3,(H:0.2,I:0.2):0.3):0.4,((J:0.2,(K:0.1,L:0.1):0.1):0.3,((M:0.2,(N:0.1,O:0.1):0.1):0.2,(P:0.2,(Q:0.1,R:0.1):0.1):0.2):0.3):0.4);

**Multiple Sequence Alignment**

| A | ATCGCCAACCTCGACTCAGTC |
|---|---|
| B | CCAGCAGCCTACATGATCATC |
| C | **CACGGCC**TACGCATCGAACTA |
| D | TCAGTCATGGTTTCATCTGAC |
| ⋮ | |
| R | CGCCTACCAATCGACTGACTG |
| Q | **CACGGCC**AGACATCAGTCAGT |

**A:** Yes, Taxa C and Q

**REL-B1 (RC=5)**

**Q:** Homoplasy refers to structured convergence, pairs or groups of distantly related taxa that repeatedly share the same nucleotide motifs, across many independent alignment columns more often than expected, while other taxa with similar overall sequences do not share those nucleotide motifs as consistently. Your job is to examine the entire alignment and provided tree and decide whether such structured homoplasy is likely to be present and which taxa are involved. Return your answer as: Yes/No and if Yes, list the taxa involved.

**Phylogenetic Tree (Represented in Newick tree format)**

((((A:0.2,((B:0.1,C:0.1):0.1,D:0.2):0.1):0.2, (E:0.2,(F:0.1,G:0.1):0.1):0.2):0.3,(H:0.2,I:0.2):0.3):0.4,((J:0.2,(K:0.1,L:0.1):0.1):0.3,((M:0.2,(N:0.1,O:0.1):0.1):0.2,(P:0.2,(Q:0.1,R:0.1):0.1):0.2):0.3):0.4);

**Multiple Sequence Alignment**

| A | **CACGGCC**ACCTCGACTCAGTC |
|---|---|
| B | CCAGCAGCCTACATGATCATC |
| C | **CACGGCC**TACGCATCGAACTA |
| D | TCAGTCATGGTTTCATCTGAC |
| E | CGACCAATCAGTCAAGTCAG |
| F | **CACGGCC**ACCTTGACATGGAC |
| K | **CACGGCC**TGGTTTCATCTGAC |
| R | CGCCTACCAATCGACTGACTG |
| Q | **CACGGCC**AGACATCAGTCAGT |

**A:** Yes, Taxa A, C, F, K and Q

## Chemistry

**REL-C1 (RC=2)**

**Q:** Is this list of molecules a set of *constitutional isomers* (same molecular formula, different connectivity)?
1. CC(C)(C)OOO, 2. CC(CO)COO, 3. CC(C)OOCO, 4. COC(C)(C)OO, 5. C=C(CBr)CBr

**Molecules**

**A:** No

**REL-C2 (RC=2)**

**Q:** Given the following list of SMILES, what is the largest *connected* common chemical motif (maximum common substructure) present in every molecule?
1. CC(N)(Cc1ccc(O)c(O)c1)C(=O)O
2. CCOC(=O)C(C)(N)Cc1ccc(O)c(O)c1
3. CC(Cc1ccc(O)c(O)c1)C1(NN)C(=O)O
4. NC(C(=O)O)C(O)c1ccc(O)c(O)c1
5. NC(Cc1ccc(OP(=O)(O)O)c(O)c1)C(=O)O

**Molecules**

**A:** NC(Cc1ccc(O)c(O)c1)C(=O)O

**Maximum common substructure**

**REL-C3 (RC≥5)**

**Q:** Given the following list of constitutional isomers, complete the set by identifying the missing constitutional isomers.
1. CC(C)N(C)N, 2. CNC(C)(C)N, 3. CCC(C)(N)N, 4. CC(N)CN, 5. CNCC(C)N

**Molecules**

**A:** CC(C)(C)NN, CC(C)(N)CN, CC(C)(N)CN, CC(C)CNN, CC(C)NCN, CC(N)C(C)N, CC(N)CCN, CC(N)N(C)C, CCC(C)NN, CCC(N)CN, CCC(N)NC, CCCC(N)N, CCCCNN, CCCN(C)N, CCCNCN, CCCNNC, CCN(C)CN, CCN(C)NC, CCN(N)CC, CCNC(C)N, CCNCCN, CCNCNC, CCNN(C)C, CCNNCC, CN(C)CCN, CN(C)N(C)C, CNC(C)CN, CNC(C)NC, CNCCCN, CNCCNC, CNCN(C)C, CNNC(C)C, NCCCCN

*Figure 2.* **REL** evaluates relational reasoning across algebraic, biological, and chemical domains.

**RPM₁ (RC = 1)**

a)   b)   **c)**

**RPM₂ (RC = 2)**

a)   **b)**   c)

**RPM₃ (RC = 4)**

**a)**   b)   c)

*Figure 3.* Three examples of Raven's Progressive Matrices with increasing relational complexity. The answers are shown in bold.

**Average).** Each entry is the sum of the same 4 predecessors along the $x$-, $y$-, and $z$-axis. RC = 4. **A6 (5-Moving Average).** Same as previous, but with 5 predecessors. RC = 5. **A7 (Neighborhood Sum).** Each entry of the RPT is the sum of its neighbors modulo 7. RC = 6 in a $3 \times 3 \times 3$ tensor.

We use the same setup as in the original vision-based RPM task, the model needs to determine the missing value based on the given matrix or tensor. We detail the generation of

REL-A in Appendix A.1. The resulting **REL**-A dataset consists of 3,500 RPMs/RPTs in total, but the synthetic dataset generators introduced here allow for the construction of potentially many more **REL**-A type questions with various sizes and value ranges.

### 4.2. Relational Reasoning in Biology (REL-B)

There exists many benchmarks for biological sequences, including ProteinGym (Notin et al., 2023) that evaluates whether a model can predict the effect of individual variants in protein sequences, DNALongBench (Cheng et al., 2025) that evaluates whether a model can predict the long-range effects of genomic elements, and TAPE (Rao et al., 2019) and PEER (Xu et al., 2022) with diverse tasks such as protein-protein interaction and protein localization prediction. These benchmarks typically evaluate tasks defined for individual sequences or sequence pairs (Rong et al., 2025; Gao et al., 2025; Ye et al., 2024).

However, many biological inferences fundamentally require

relational comparisons across multiple sequences conditioned on their evolutionary history, such as detecting convergent evolution or homoplasy across organisms (Wake et al., 2011), where similar motifs arise independently in distinct lineages (Appendix B.2). To evaluate a model's ability to detect homoplasy, we provide a model with a multiple sequence alignment (MSA) and the corresponding phylogenetic tree and ask it to: (1) decide whether homoplasy is present, and (2) identify the taxa participating in the homoplastic motif (Fig. 4). Solving the task requires jointly (a) localizing a shared motif across sequences and (b) verifying from the tree that the motif spans evolutionarily distinct lineages rather than a single recent clade.

Here, $RC = N_{ht}$, where $N_{ht}$ is the number of homoplastic taxa, as the model must maintain in its memory the current position of the taxa in relation to the other homoplastic taxa in the tree. Taxa denote taxonomic groups of any rank, such as species, families, or classes.

To systematically evaluate this capability, we construct a synthetic dataset generator parameterized by four variables: (1) the number of homoplastic taxa $N_{ht}$, (2) the number of leaves in the phylogenetic tree $N_{\text{leaves}}$, (3) the sequence length $L_{\text{seq}}$, and (4) the length of the conserved motif $L_{\text{motif}}$ (Figure 4). This construction enables the generation of a large and diverse set of questions spanning a wide range of difficulty regimes, while preserving a known ground truth for both homoplasy presence and taxon identity. In particular, the generator allows us to scale the dataset combinatorially across parameter settings, producing many distinct MSAs and trees without manual curation. Notably, **REL**-B1 is scalable to various levels of RC as we can adjust the number of homoplastic taxa.

For the evaluations below we generated 2,600 questions. Further details on question generation and parameters used are available in Appendix A.3 and additional biology questions are also available in Appendix A.3.2 (**REL**-B2).

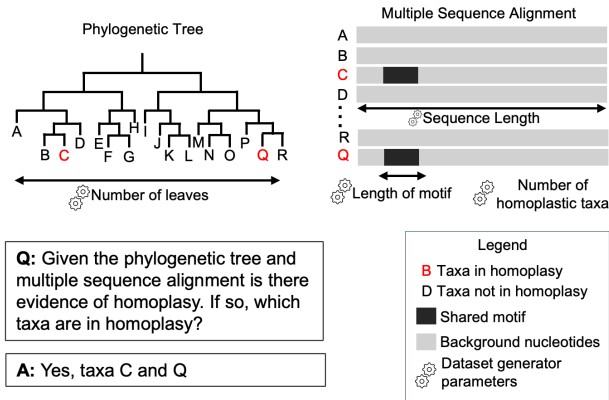

*Figure 4.* From provided parameters, we generate a phylogenetic tree, alignment, inject shared motifs, and ask the model to use this alignment and tree to identify the taxa that are in homoplasy.

## 4.3. Relational Reasoning in Chemistry (REL-C)

Relational reasoning is key to understanding molecular function and the vast chemical space (Bemis & Murcko, 1996). This capability is central to chemical library design (Hajduk et al., 2011) and the analysis of structure-activity relationships (Guha, 2013) in drug discovery. Chemically meaningful inferences rely on shared functional motifs between molecules. For example, molecules which share an aromatic ring, such as a phenyl group, often exhibit similar stacking interactions and hydrophobic properties. In contrast, current approaches focus on evaluating molecular representations and editing of one molecule or combining molecules in a reaction (Fang et al., 2023; Jang et al., 2025; Runcie et al., 2025; Li et al., 2025; Liu et al., 2025b). Here, we introduce three tasks where the entities consist of molecules and tasks involve resolving relations of increasing complexity between the molecules.

**C1: Constitutional isomer set classification.** This question evaluates binary classification of whether a set of molecules forms a constitutional isomer set, molecules that have the same molecular formula but different bond connectivity. Here, $RC = 2$ as the model must maintain the following to complete the task: (1) the current shared molecular formula, and (2) the formula of the current molecule in the set to iteratively confirm if all molecules share the same molecular formula. For this task, we construct isomers by sampling molecular formulae spanning $C_{3-9}$ with heteroatoms (O, N, S, F, Cl, Br) and various degrees of unsaturation. For "No" instances, we sample $n-1$ molecules from one formula and 1 molecule from a different formula. In total 1,000 questions were curated for this task, with 100 questions for each $N_{\text{molecules}} \in \{5, 10, 15, 20, 25, 30, 35, 40, 45, 50\}$ (Appendix A.4.1). For **REL**-C1, the task completion rate is defined as the accuracy of the responses.

**C2: Largest continuous common chemical motif.** C2 evaluates the ability to identify the maximum common substructure (MCS) shared across a set of molecules. We construct instances by sampling similar molecules from drug-like compounds sampled from ChEMBL (Mendez et al., 2019). The ground-truth MCS is required to contain at least 8 atoms, is connected, and maximizes the number of bonds. The bottleneck is binary, as the model needs to maintain the current largest common substructure and update it with each new molecule in the set. In total, we generate 1,016 questions (Appendix A.4.2) with $N_{\text{molecules}} \in \{5, 10, 15, 20, 25, 30, 35, 40, 45, 50\}$, the number of questions for each number of molecules is provided in Table A.3.

While both **REL**-C1 and **REL**-C2 have binary relational complexity, the operand complexity between a pair of molecules in **REL**-C2 is more challenging than in **REL**-C1. In **REL**-C1, two molecules are related if they share the

same chemical formulae, which is obtained by counting the number of atoms of each element. Conversely, the relation between two molecules in **REL**-C2 is determined by the largest common substructure.

Predictions are evaluated using a bidirectional substructure metric. We first compute the fraction of examples where the prediction is a substructure of the ground truth ($S_{\text{pred} \subseteq \text{true}}$) and the fraction where the ground truth is a substructure of the prediction ($S_{\text{true} \subseteq \text{pred}}$). The final metric is:

$$\text{IsSubstructure} = \frac{1}{2}\left(S_{\text{pred} \subseteq \text{true}} + S_{\text{true} \subseteq \text{pred}}\right) \qquad (1)$$

This captures both precision (avoiding extraneous atoms) and completeness (including all correct atoms), providing a more nuanced evaluation than binary accuracy for the maximum common substructure task.

**C3: Missing isomer completion.** C3 evaluates the ability to complete a constitutional isomer family given a partial set of observed molecules. To answer this question, the model must infer the full space of valid constitutional isomers implied by the shared molecular formula and identify which of these structures are not present in the observed set. Unlike C1 and C2, this task cannot be reduced to a serial binary update: determining whether a candidate isomer is missing requires simultaneously binding (1) the shared molecular formula, (2) the molecules in the isomer family, (3) the subset of isomers already observed. The relational complexity of C3 arises from the need to simultaneously bind multiple independently varying sources, including the full isomer space of size $N_{\text{isomers}}$ and an observed subset of size $N_{\text{observed}}$. We generate a total of 1,000 questions, where the average number of isomers to be identified is 29 (Appendix A.4.3). The task completion rate is given by the recall of missing isomers.

Across **REL**-C1, C2, and C3, we generate a total of 3,016 questions. Because **REL** is a generative framework, additional questions at fixed levels of relational complexity can be sampled from the molecule bank. Additional chemistry questions are also available in Appendix A.4.4 (**REL**-C4).

## 5. Experiments

We benchmark Claude Opus 4.5, Gemini 3 Pro Preview, and GPT 5.2 on questions from **REL**.

### 5.1. Algebra Tasks

**Model evaluation.** We provide RPMs to the model using the format "$[a_{1,1}, ..., a_{1,n}] \mid ... \mid [a_{n,1}, ..., a_{n,n}]$", where $[a_{i,1}, ..., a_{i,n}]$ is the $i$-th row of an $n \times n$ input. The location of the missing value that the mode is asked to provide is marked with "?", as in Fig. 2. In accordance with the original cognitive tasks given to adolescents (Carpenter et al., 1990) and with other works using RPMs to test machine

learning models (Camposampiero et al., 2025a; Hersche et al., 2025), we give the model multiple (8) different possible answer values to choose from, only one of which is correct, so trivial accuracy is 12.5%.

**Results.** We present our evaluation results in Fig. 5. All three models solve the tasks with low RC (**REL**-A1, where RC=1 and **REL**-A2, where RC=2) almost perfectly, with accuracy reaching 91% even on the largest $30 \times 30$ RPMs. On tasks where RC scales with the size of the input (**REL**-A3 and **REL**-A4), all three models struggle with larger RPM: Claude and Gemini drop to trivial accuracy (around 12%) on **REL**-A3 $30 \times 30$; while GPT-5.2's accuracy drops by nearly 40%. On **REL**-A4, this trend is even more pronounced, with only GPT-5.2 achieving non-trivial accuracy (21%) on $9 \times 9$ inputs. All three models fail on larger inputs.

Our RPT results show the opposite trend: **REL**-A5, **REL**-A6, and **REL**-A7 have, by design, a higher RC (4-6) on small inputs than the RPM tasks (1-3), but unlike with **REL**-A3 and **REL**-A4, their RC does not increase with the size of the input. As such, we find that more data, i.e. larger inputs, result in better model performance on **REL**-A5 and **REL**-A6 across all three models (for 56-64% to 77-87% on **REL**-A5 and from 41-50% to 53-64% on REL-A6). Only **REL**-A7 with its high initial RC (6) remains unsolvable for all models tested here, irrespective of input size, with an average accuracy of around 12%.

### 5.2. Biology Tasks

**Model evaluation.** A question is scored as correct only if the model correctly detects the presence or absence of homoplasy and, when present, exactly identifies the set of homoplastic taxa. All other outcomes are counted as incorrect.

**Results.** Increasing the number of homoplastic taxa leads to a sharp decrease in model performance from 35% for $N_{ht} = 4$ to 1% for $N_{ht} = 25$ when averaged across models (Fig. 6). To assess whether this effect could be explained by alternative explanations we examined four other factors: motif ratio (the ratio of motif length to sequence length), sequence length, average pairwise distance between homoplastic taxa, and prompt length (Fig. 7). Increasing the motif ratio from 10-12.5% to 25-30% increased performance from 12.6% to 25.1%. Increasing the sequence length from 500 to 900 increased performance from 17.8% to 19.6%. Increasing distance from 5-6 to greater than 15 decreased performance from 10.2% to 3.2%. While these factors influence performance, none exhibit a comparably large effect across their full range (Fig. A.1).

To quantify the independent contribution of each factor, we fit separate multivariate regression models for each LLM and measured the unique share of explainable variance con-

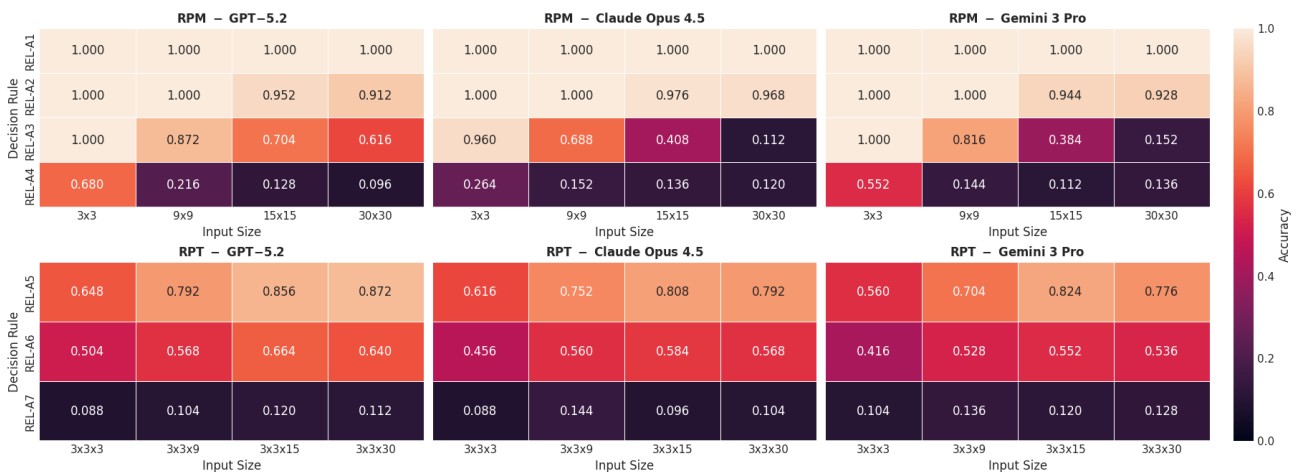

*Figure 5.* Model performance on **REL**-A tasks. RPMs at the top, with RPTs below. The models are given 8 answer choices, so trivial accuracy is 12.5%. All three model perform well on tasks with low RC (**REL**-A1 and **REL**-A2, top two rows), but struggle once RC increases: on **REL**-A3 and **REL**-A4, where RC increases with input size, performance drops by as much as 80%. RPTs (**REL**-A5, **REL**-A6, and **REL**-A7), which always have a higher RC, independent of input size, are challenging to impossible for all three models.

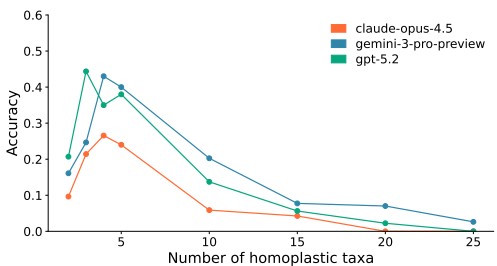

*Figure 6.* Performance decreases with increasing RC controlled by increasing the number of homoplastic taxa increases in **REL**-B1.

tributed by each variable. Across all three models, RC explains the largest share of explainable variance: 24% of explainable variance for Claude, 32% for Gemini, and 44% for GPT. In contrast, the next strongest factor, motif ratio for Claude, prompt length for Gemini, and distance between taxa for GPT, explains 7%, 17%, and 6% of explainable variance for Claude, Gemini, and GPT, respectively (Fig. 7). To assess whether correlations among predictors influenced these estimates, we performed a collinearity analysis using generalized variance inflation factors (GVIF). We found no problematic collinearity for the key variables: number of homoplastic taxa (1.17), distance bin (1.18), and motif ratio bin (1.30). Sequence length and prompt bin were higher (7.23 and 5.37), which is expected because longer sequences mechanically induce longer prompts. Overall, these results suggest that RC is the dominant driver of model performance, and that the degradation observed on **REL**-B1 is not explained by the other measured factors.

### 5.3. Chemistry Tasks

**Model evaluation.** All molecules are provided to the models as canonicalized SMILES. Model responses are evaluated by canonicalizing both predicted and ground-truth

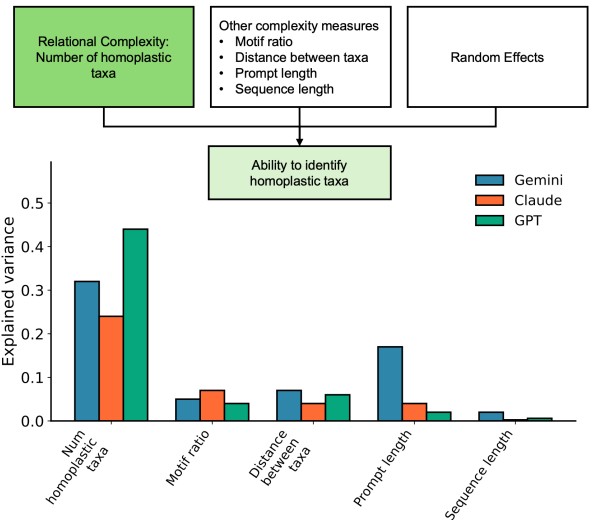

*Figure 7.* **Top:** Schema of variables in multivariate regression. **Bottom:** Explained variance of performance on **REL**-B1 across five measures of complexity. RC, which is number of homoplastic taxa, explains the most variance.

SMILES strings and comparing the canonical forms for exact match, ensuring that chemically equivalent SMILES representations are treated as correct. Example prompts of the three tasks are provided in Appendix E.3.

**Results.** We find that across our chemistry tasks, as RC increases, the task completion rate decreases (Fig. 8). **REL**-C1 has the highest average task completion rate at 65.7%, compared to **REL**-C2, which has an average task completion rate of 38.1%. While these two tasks share the same RC, OC of **REL**-C2 is higher than that of **REL**-C1. Finally, the task with the lowest task-completion rate is **REL**-C3 at 26.0%.

Both **REL**-C1 and **REL**-C2 have fixed RC with RC=2

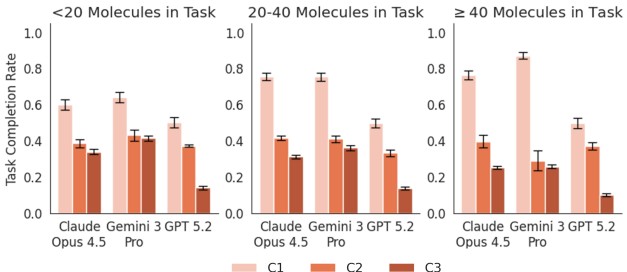

*Figure 8.* Task completion rate decreases as from C1 (RC=2 and OC easy) to C2 (RC=2 and OC medium) to C3 (RC=$N_{\text{isomers}}$, $N_{\text{observed}}$ and OC hard). This observation holds across different number of molecules in the task.

across $N_{\text{molecules}}$. We observe that **REL**-C1 increases in accuracy from 56.0% at 5 molecules to 71.0% at 50 molecules. In **REL**-C2, the task completion rate remains relatively stable for averaging at 39.2% ± 2.6% for $5 \leq N_{\text{molecules}} < 50$ (Fig. 9L). This affirms that the number of entities does not affect performance until $N_{\text{molecules}}$ becomes very large. OC drives the decreased task completion rate between **REL**-C1 and **REL**-C2, as determining the MCS is more challenging than determining if molecules share the same molecular formulae. The OC of determining MCS also increases as the number of atoms in the input molecules increases, which is associated with a drop in performance (Fig. 9R).

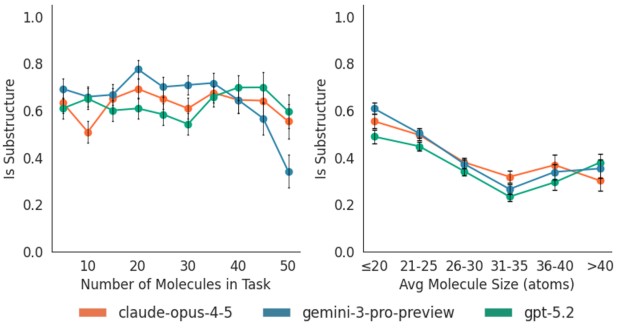

*Figure 9.* Task completion rate on **REL**-C2 evaluated with Is Substructure. For this task, RC is fixed at 2 in **Left:** increasing $N_{\text{molecules}}$ does not have an effect on performance until $N_{\text{molecules}} = 50$, in **Right:** increasing the molecule size increases OC and leads to decreased IsSubstructure rate.

Finally, for **REL**-C3, RC increases with both $N_{\text{isomers}}$ and $N_{\text{missing}} = N_{\text{isomers}} - N_{\text{observed}}$. We observe that increasing both sources of RC decreases performance across all three models, with recall decreasing from 30.0% for $N_{\text{isomers}} = 5$ to 21.2% for $N_{\text{isomers}} = 50$ (Fig. 10L). In addition to evaluating task completion rate as recall, we also report precision and F1 which show similar decreasing performance with increasing RC in Table A.6.

### 5.4. Effects of Inference-Time Interventions

**Test-Time Compute.** We analyze how test-time compute affects performance on **REL**-A and **REL**-C. In Table A.8

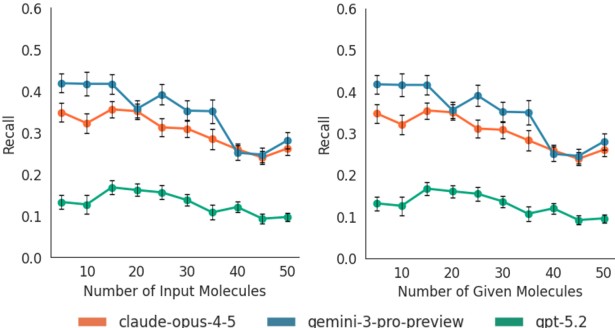

*Figure 10.* Task completion rate on **REL**-C3 evaluated with recall, RC depends on both $N_{\text{observed}}$ and $N_{\text{missing}}$. **Left:** As RC increases with $N_{\text{observed}}$ recall decreases. **Right:** As RC increases with $N_{\text{missing}}$ recall decreases.

we show results for REL-A4 and REL-A5 inputs where models are given a maximum token threshold of 4,096, 8,192, and 16,384 tokens. This tends to increase accuracy by 2-3% in accuracy, but does not close the gap to performance on tasks with lower RC. In Fig. A.5 we also show results for **REL**-C1,C2,C3 where models are again given 4,096, 8,192, and 16,384 tokens. On average, we observe a 0.4% change on average in task completion rate across the three models. With increased test-time compute, we still observe that higher RC leads to worse performance.

**In-Context Learning.** We run a variation of the task with one-shot in-context learning for 10% of our tasks in **REL**-C. At $N_{\text{molecules}} < 20$ we see that in-context learning leads to a boost in performance, however relationships between tasks remain unchanged with C1 at 77.0% (+6.6%), followed by C2 at 43.3% (+3.4%), and finally C3 at 32.7% (+6.0%). While in context learning improves performance, we observe the same outcome where as RC increases between questions, task completion rate decreases. We provide additional results in Appendix D.

**Tool Use.** For **REL**-C3, we evaluated whether access to tools improved performance by providing the model with RDKit, a standard cheminformatics toolkit for parsing molecular SMILES, for all questions. Performance remained poor overall, with low mean recall (0.094), and continued to decline as the number of input molecules increased: recall was 0.109 (0.009) for 5–20 molecules, 0.094 (0.006) for 20–40 molecules, and 0.079 (0.005) for $\geq 40$ molecules (Table A.6). This suggests that the degradation is not eliminated by externalizing molecular parsing and chemistry operations.

## 6. Discussion

Across **REL**, our experiments reveal a consistent bottleneck: model performance tracks RC more reliably than traditional proxies such as input size or the number of entities. In

**REL**-A, models solve low-RC RPM rules nearly perfectly, but accuracy collapses once the governing rule requires higher-arity integration. In **REL**-B1, RC, controlled by the number of homoplastic taxa, explains most of the variance in performance degradation. Finally, in chemistry, increasing RC across **REL**-C1, **REL**-C2, and **REL**-C3 induces a uniform decline in task completion rates.

These findings have two implications. First, many benchmark improvements that appear to reflect better reasoning may instead arise from gains along axes orthogonal to relational integration, and re-evaluating benchmarks through the lens of RC may therefore yield more diagnostic comparisons across models and inference settings. Second, the observed failure regime appears persistent: allocating additional test-time thinking provides limited benefit on high-RC instances, and several tasks remain effectively unsolved across all evaluated models.

Our study has several limitations: multiple-choice evaluation may obscure finer-grained reasoning failures, context-length constraints lead to invalid responses in some settings, and our tasks remain relatively synthetic. Addressing these limitations is an important direction for future work. Going forward, we aim to expand **REL** to more naturalistic relational settings and to explore approaches to improve target higher RC reasoning. With **REL**, we provide a framework for evaluating model performance based on their ability to reliably compose many relations simultaneously.

## Impact Statement

This paper aims to advance machine learning by characterizing how large language model (LLM) performance varies with relational complexity, situations where solving a problem requires representing and manipulating multiple relations simultaneously. As LLMs are increasingly deployed in real-world settings, understanding systematic failure modes is important for safer and more appropriate use. Our results suggest that increasing relational complexity can lead to substantial performance degradation, which may inform practitioners about when additional safeguards, alternative methods, or human oversight are warranted, especially in higher-stakes applications.

## Acknowledgments

L.F. and A.F. are supported by the Kempner Graduate Fellowship at Harvard University. Y.E. is supported by the Eric and Wendy Schmidt Center at Broad Institute. We gratefully acknowledge the support by NSF CAREER Award 2339524, ARPA-H Biomedical Data Fabric (BDF) Toolbox Program, Amazon Faculty Research, Google Research Scholar Program, AstraZeneca Research, GlaxoSmithKline Award, Roche Alliance with Distinguished Scientists (ROADS) Program, Sanofi iDEA-iTECH Award, Boehringer Ingelheim Award, Merck Award, Optum AI Research Collaboration Award, Pfizer Research, Gates Foundation (INV-079038), Chan Zuckerberg Initiative, Collaborative Center for XDP at Massachusetts General Hospital, John and Virginia Kaneb Fellowship at Harvard Medical School, Biswas Computational Biology Initiative in partnership with the Milken Institute, Harvard Medical School Dean's Innovation Fund for the Use of Artificial Intelligence, and the Kempner Institute for the Study of Natural and Artificial Intelligence at Harvard University. Any opinions, findings, conclusions or recommendations expressed in this material are those of the authors and do not necessarily reflect the views of the funders.

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

# A. Details on the Construction of REL Tasks

## A.1. REL-Algebra

For each problem size and ground rule, we generate 125 distinct RPMs/ RPTs.

- For **REL**-A1, we sample an integer value randomly from a predefined domain.

- For **REL**-A2, we randomly choose "plus" or "minus" for the progression and then sample an integer increment, as well as initial values for each row uniformly from a predefined domain.

- For **REL**-A2, we randomly choose "plus" or "minus" for the progression and then sample an integer increment, as well as initial values for each row uniformly from a predefined domain.

- For **REL**-A4, we sample "plus" or "minus" $n-1$ times and then populate the first $n-1$ columns with integers from a predefined domain, which determines the last column.

- The RPTs for **REL**-A5 and **REL**-A6 are both generated using randomly sampled values for the first $3 \times 3$ values of the RPT, and the moving average rule then determines the rest of the tensor.

- Finally, generating **REL**-A7 RPTs requires a more involved algorithm, which we elaborate on below.

## A.2. Construction of REL-A7

**Problem Definition.** Given parameters:

- Grid size: $n \times n$ (typically $n = 3$)

- Number of slices: $K$ (depth of tensor)

- Maximum value: maxval

- Prime modulus: $p$

Generate a 3D tensor $A \in \mathbb{Z}p^{n \times n \times K}$ such that for each slice $k \in \{0, 1, \ldots, K-1\}$ and each cell $(i, j)$, the value $Ai, j, k$ satisfies the neighborhood sum constraint modulo $p$ using 4-connected neighbors (cardinal directions).

**Neighborhood Definition.** For a cell at position $(i, j)$ in slice $k$, define the 4-connected neighborhood set $\mathcal{N}_{i,j}$:

$$\mathcal{N}_{i,j} = \{(i-1 \bmod n, j), (i+1 \bmod n, j), (i, j-1 \bmod n), (i, j+1 \bmod n)\} \tag{2}$$

**Mathematical Formulation.** For each slice $k \in \{0, 1, \ldots, K-1\}$ and cell $(i, j)$, the constraint is:

$$A_{i,j,k} \equiv \sum_{(i',j') \in \mathcal{N}i,j} A_{i',j',k} \pmod{p} \tag{3}$$

---

**Algorithm 1** Self-Consistent NeighborhoodSum RPT Generation

---

**Require:** $n, K, \text{maxval}, p$

**Ensure:** Tensor $A \in \mathbb{Z}_p^{n \times n \times K}$ where every cell satisfies the neighborhood constraint, missing cell position $(k, i, j)$, target value $t$, and answer candidates

1: Initialize $A$ randomly: $A_{i,j,k} \leftarrow \text{RandomInt}(0, \text{maxval})$ for all $i, j, k$
2: converged $\leftarrow$ False
3: **while** not converged **do**
4:     converged $\leftarrow$ True
5:     **for** $k = 0$ to $K - 1$ **do**
6:         **for** $i = 0$ to $n - 1$ **do**
7:             **for** $j = 0$ to $n - 1$ **do**
8:                 sum $\leftarrow 0$
9:                 **for** $(i', j') \in \mathcal{N}_{i,j}$ **do**
10:                     sum $\leftarrow$ sum $+ A_{i',j',k}$
11:                 **end for**
12:                 new_val $\leftarrow$ sum $\bmod p$
13:                 **if** $A_{i,j,k} \neq$ new_val **then**
14:                     $A_{i,j,k} \leftarrow$ new_val
15:                     converged $\leftarrow$ False
16:                 **end if**
17:             **end for**
18:         **end for**
19:     **end for**
20: **end while**
21: **Select missing cell**
22: $(k^{\cdot}i^{\cdot}j^*) \leftarrow \text{RandomChoice}(\{(k, i, j) : k \in \{0, \ldots, K - 1\}, i, j \in \{0, \ldots, n - 1\}\})$
23: $t^* \leftarrow A_{k^{\cdot}i^{\cdot}j^*}$
24: **Generate answer candidates**
25: candidates $\leftarrow [t^*]$ {Correct answer}
26: **while** $|\text{candidates}| < 8$ **do**
27:     $d \leftarrow \text{RandomDistractor}()$ {Generate distractor value}
28:     **if** $d \notin$ candidates **then**
29:         candidates.append($d$)
30:     **end if**
31: **end while**
32: candidates $\leftarrow \text{Shuffle}(\text{candidates})$
    **Return** $(A, (k, i, j), t^*, \text{candidates})$

---

## A.3. REL-Biology

### A.3.1. **REL**-B1: IDENTIFYING HOMOPLASTIC TAXA

Each dataset instance is generated as follows:

1. **Sample a random tree.** Draw a random Newick tree with $n_{\text{leaves}}$ taxa.

2. **Simulate a baseline alignment.** Using Pyvolve, simulate a nucleotide alignment of length $l_{\text{seq}}$ under a standard substitution model.

3. **Inject tree-aware convergent blocks.** Inject a motif of length $l_{\text{motif}}$ by enforcing a shared motif across taxa that are distant on the tree:

   - Select $n_{ht}$ leaves whose pairwise *topological distance* (the number of edges along the unique path between two leaves) is at least 3.
   - For a randomly chosen contiguous block of $l_{\text{motif}}$ columns, overwrite the nucleotides for the selected taxa with the same base (or motif), inducing a structured convergence signal spanning multiple columns.

We generate datasets starting from a baseline configuration with $n_{\text{leaves}} = 50$, $l_{\text{seq}} = 1000$, $l_{\text{motif}} = 50$, and $n_{ht} = 2$, and then vary one parameter at a time. Specifically, we consider $n_{ht} \in \{2, 3, 4, 5, 10, 15, 20, 25\}$, $n_{\text{leaves}} \in \{20, 30, 40, 100, 1000\}$, $l_{\text{seq}} \in \{200, 300, 500, 1000, 2000\}$, and $l_{\text{motif}} \in \{3, 4, 5, 30, 40, 50\}$, we also resulting in a total of 2,600 questions. Across the three evaluated LLMs, this yields 7,800 API calls.

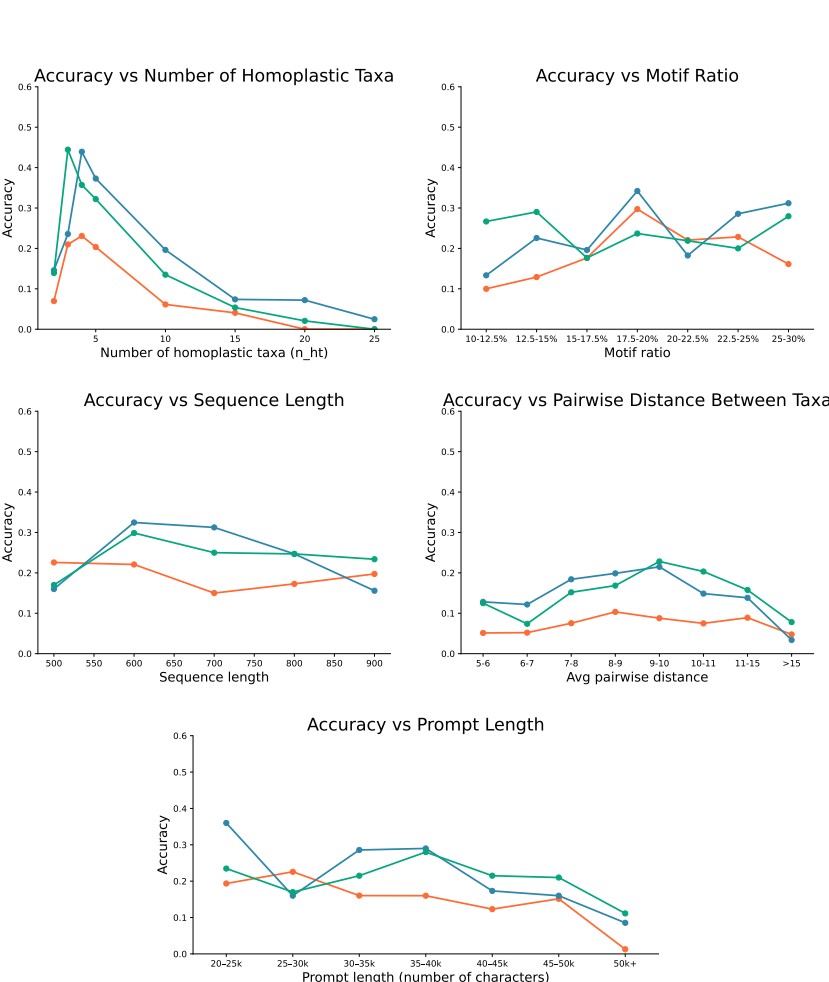

*Figure A.1.* Accuracy as a function of the number of homoplastic taxa, motif ratio, sequence length, average pairwise distance between taxa, and prompt length for **REL**-B1.

### A.3.2. **REL**-B2: UNCOVERING EPISTATIC STRUCTURE

Many biological phenotypes are shaped not only by the marginal effects of individual mutations, but by how mutations interact jointly through epistasis (Mackay, 2014). To evaluate this form of biological relational reasoning, we introduce a task based on experimentally measured combinatorial protein fitness landscapes. We use local landscapes derived from antibody binding data (HA and HA2), a GFP mutational landscape, and four-residue local landscapes from GB1 and TrpB (Wu et al., 2016; Phillips et al., 2021; Poelwijk et al., 2019). In contrast to standard variant-effect prediction benchmarks, which typically ask for the effect of a single sequence or sequence pair, this task requires inferring the *interaction structure* of a local fitness landscape from many measured variants at once.

For each example, we select a set of $k$ focal mutational variables and fix the remaining assayed positions as a background $b$. This defines a local fitness function:

$$f_b : \{0, 1\}^k \to \mathbb{R},$$

where $x_i = 0$ denotes the reference state of focal mutation $i$, $x_i = 1$ denotes the mutant state, and $f_b(x)$ is the experimentally measured fitness of the corresponding sequence. For GFP, HA, and HA2, we sample $k$ assayed positions and randomly assign reference or mutant states to the non-focal assayed positions to define the background. We then enumerate all $2^k$ focal combinations and retain the example only if every combination is present in the measured dataset. For GB1 and TrpB, which are represented as measured four-site landscapes, we select $k$ focal positions from the four assayed residues and treat the remaining assayed positions as a fixed background.

To extract the latent interaction structure of each local landscape, we compute epistatic coefficients using the unnormalized Walsh–Hadamard transform, equivalently the Möbius transform on the Boolean hypercube (Poelwijk et al., 2016). For each subset $S \subseteq \{1, \ldots, k\}$ with $|S| \geq 2$, we define:

$$W_b(S) = \sum_{T \subseteq S} (-1)^{|S|-|T|} f_b(\mathbf{1}_T),$$

where $\mathbf{1}_T \in \{0, 1\}^k$ denotes the binary genotype in which exactly the focal mutations in $T$ are present and all others are absent. Intuitively, $W_b(S)$ measures the component of the landscape that cannot be explained by lower-order additive effects alone. For example, the pairwise coefficient is as follows:

$$W_b(\{i, j\}) = f_{11} - f_{10} - f_{01} + f_{00}$$

measures deviation from additivity between mutations $i$ and $j$, while the third-order coefficient is defined:

$$W_b(\{i, j, \ell\}) = f_{111} - f_{110} - f_{101} - f_{011} + f_{100} + f_{010} + f_{001} - f_{000}$$

captures irreducible three-way dependence beyond all single and pairwise terms.

We use these coefficients to assign each local landscape to a coarse epistatic structure class. Let:

$$\Delta_b = \max_x f_b(x) - \min_x f_b(x)$$

denote the dynamic range of the local landscape, and define a salience threshold:

$$\tau_b = 0.12 \, \Delta_b.$$

Only coefficients with magnitude exceeding $\tau_b$ are treated as structurally meaningful. We first identify the dominant interacting pair as:

$$S_2^\star = \arg \max_{|S|=2} |W_b(S)|.$$

Among all third-order coefficients containing this pair, we then identify the strongest associated trio as:

$$S_3^\star = \arg \max_{|S|=3, \, S_2^\star \subseteq S} |W_b(S)|.$$

To quantify whether a focal mutation behaves approximately independently of the others, we define its interaction score as:

$$I_b(i) = \max_{S \ni i, \, |S| \geq 2} |W_b(S)|.$$

A mutation is treated as approximately independent when $I_b(i) < \tau_b$.

These statistics are mapped to an interpretable structural label (Sailer & Harms, 2017). For **REL**-B2 instances with $k = 2$, we assign one of three classes: positive epistasis, negative epistasis, or approximate independence, according to the sign and magnitude of the single pairwise coefficient. For $k = 3$, we distinguish whether the landscape is best explained by (i) a dominant pairwise interaction with the third mutation acting approximately independently, (ii) a three-way modulation of the dominant pair, or (iii) a hub-like pattern in which one mutation participates in multiple pairwise interactions without a strong irreducible three-way term. For $k = 4$, we additionally distinguish whether the fourth mutation is approximately independent, yielding classes that correspond to a dominant pair with independent remainder, trio modulation with one independent mutation, hub-like structure with one independent mutation, or broader coupling across all four focal mutations. For $k \geq 5$, we extend the same logic to a dominant-structure summary: we identify the leading pair, the strongest associated trio, and whether the remaining mutations are mostly independent, form an additional interacting group, or are broadly coupled. Accordingly, for larger $k$ the benchmark captures the *dominant* epistatic structure of the local landscape rather than an exhaustive taxonomy of all higher-order interactions.

The model does not observe Walsh coefficients directly. Instead, it is shown the complete measured fitness table over all $2^k$ focal combinations together with multiple-choice natural-language explanations of the latent interaction structure. The correct answer and distractor options are generated from the coefficient-based structural analysis. For example, a dominant

*Table A.1.* Performance across relational complexity (RC) levels for each dataset in **REL**-B2 task for GPT-5.2.

| Dataset | RC-2 | RC-3 | RC-4 | RC-5 | RC-6 |
|---|---|---|---|---|---|
| **GFP** | 81.4% | 27.5% | 46.3% | 13.2% | 11.8% |
| **HA** | 81.4% | 34.3% | 47.1% | 16.2% | 8.0% |
| **HA2** | 83.3% | 35.3% | 40.4% | 14.0% | 6.6% |
| **GB1** | 78.4% | 15.7% | – | – | – |
| **TrpB** | 76.5% | 21.6% | – | – | – |
| **Chance** | 33% | 25% | 25% | 25% | 25% |

negative pairwise interaction may be verbalized as two mutations being "harmful together," whereas a salient third-order term involving a third mutation may be verbalized as that mutation "modulating" or "rescuing" the pairwise effect depending on the sign and context. Solving the task therefore requires integrating evidence across many mutational backgrounds to infer the best global explanation of the landscape, rather than reading off any single local effect.

We define the relational complexity of this task as:

$$\mathrm{RC} = k,$$

where $k$ is the number of focal mutations whose context-dependent effects must be jointly represented to infer the correct structural explanation of the local fitness landscape. This follows the general REL definition of RC as the number of independently varying sources that must be bound simultaneously to carry out the required reasoning step.

### A.4. REL-Chemistry

#### A.4.1. **REL**-C1

For the candidate formulas spanning $C_{3-9}$ with various heteroatoms (O, N, S, F, Cl, Br) and degrees of unsaturation, we generate isomers using the Surge structure enumeration tool (McKay et al., 2022). After generation, we filter to retain formulas with 5-100 isomers to ensure both tractability and sufficient sampling diversity.

*Table A.2.* Details for **REL**-C1. Below double bond equivalent (DBE) measures the degree of unsaturation in the formula.

| $N_{\mathrm{molecules}}$ | 5 | 10 | 15 | 20 | 25 | 30 | 35 | 40 | 45 | 50 |
|---|---|---|---|---|---|---|---|---|---|---|
| # Questions | 100 | 100 | 100 | 100 | 100 | 100 | 100 | 100 | 100 | 100 |
| Avg. Atoms | 5.75 | 5.78 | 5.94 | 6.10 | 6.20 | 6.10 | 6.34 | 6.21 | 6.15 | 6.32 |
| Avg. DBE | 4.31 | 4.54 | 4.61 | 4.68 | 4.78 | 4.80 | 4.79 | 5.35 | 5.31 | 5.47 |

#### A.4.2. **REL**-C2

To select molecules with real-world relevance, molecules in this task are sampled from phase 1 to 4 drug molecules in ChEMBL (Mendez et al., 2019). We first construct a diverse molecular bank by filtering molecules to contain 15-60 heavy atoms and applying a greedy diversity selection algorithm that iteratively selects molecules maximally distant (by Tanimoto distance) from those already selected resulting in 9,035 molecules. For each instance at a given $N_{\mathrm{molecules}}$, we randomly select a seed molecule and sample similar molecules from the similarity range [0.35, 0.90], ensuring structural relatedness while avoiding near-duplicates. We impose a minimum MCS size of 8 atoms, as lower thresholds lead to many trivial motifs, primarily consisting of benzene and toluene substructures. In total, we generated 1,016 questions with $N_{\mathrm{molecules}} \in \{5, 10, 15, 20, 25, 30, 35, 40, 45, 50\}$ the number of questions for each number of molecules is provided in Table A.3.

#### A.4.3. **REL**-C3

For the missing isomer completion task, we leverage the same exhaustively enumerated constitutional isomer universes used in C1. For each instance at a given $N_{\mathrm{molecules}}$, we select a molecular formula whose complete isomer universe has between 8 and 100 members. From this complete universe, we randomly sample $N_{\mathrm{molecules}}$ as the "given" set presented to the model, with the remaining molecules constituting the ground-truth answer. The average number of isomers to be identified is 29. For this task, we define the task completion rate as the recall of correct isomers.

*Table A.3.* **REL**-C2. Number of questions by number of molecules. The number of questions decreases with the number of molecules as it becomes more unlikely to sample a set of molecules with a largest common motif of at least 8 atoms.

| $N_{\text{molecules}}$ | 5 | 10 | 15 | 20 | 25 | 30 | 35 | 40 | 45 | 50 |
|---|---|---|---|---|---|---|---|---|---|---|
| # Questions | 120 | 120 | 120 | 120 | 120 | 120 | 120 | 76 | 53 | 47 |
| Avg. atoms | 30.74 | 30.75 | 27.28 | 28.18 | 27.25 | 27.39 | 26.46 | 27.42 | 27.42 | 28.45 |
| Avg. elements | 3.32 | 3.34 | 3.29 | 3.34 | 3.30 | 3.25 | 3.14 | 3.02 | 2.96 | 2.83 |
| Avg. DBE | 24.55 | 24.59 | 22.11 | 22.55 | 22.09 | 22.12 | 21.79 | 23.08 | 23.27 | 24.08 |
| Avg. rings | 3.03 | 2.97 | 2.98 | 3.00 | 3.09 | 3.07 | 3.13 | 3.32 | 3.42 | 3.81 |

*Table A.4.* Most frequent molecular substructure motifs and their occurrence counts in **REL**-C2.

| Motif (SMILES) | Count |
|---|---|
| COc1ccccc1 | 180 |
| CCc1ccccc1 | 114 |
| O=[SH](=O)c1ccccc1 | 50 |
| CCOc1ccccc1 | 17 |
| CC12C=CC(=O)C=C1CCC1C2CCC2(C)C(C=O)CCC12 | 13 |
| CC(C)NCC(O)COc1ccccc1 | 9 |
| CCCCCCOC | 5 |
| CCCOc1ccccc1 | 5 |
| CC(O)C(O)C(O)CO | 4 |
| CCN(CC)CCCN1c2ccccc2Sc2ccccc21 | 4 |

### A.4.4. **REL**-C4

Here we introduce an additional question type, **REL**-C4: Constraint satisfaction with motif selection. **REL**-C4 evaluates the ability to extract molecular substructures (motifs) from a set of molecules to satisfy a global functional group constraint. Given $N_{\text{molecules}}$ molecules and a target count $T$, the model must select one continuous motif from each molecule such that the total number of a specified functional group across all selected motifs equals $T$. We consider five functional group constraints: carboxylic acids, aromatic rings, alcohols, primary amines, and ketones. This task exhibits $\text{RC} = N_{\text{molecules}}$, as the model must jointly choose a valid motif from each molecule while satisfying a global arithmetic constraint over all selections.

We construct instances by sampling drug-like molecules and using dynamic programming to identify feasible target values. Each motif must be a valid, connected substructure containing at least 6 heavy atoms. In total, we generate 1,000 questions with $N_{\text{molecules}} \in \{5, 10, 15, 20, 25, 30, 35, 40, 45, 50\}$ (100 questions per size) across the five constraint types. For **REL**-C4, task completion requires that (1) all predicted motifs are valid SMILES, (2) each motif is a substructure of its parent molecule, (3) all motifs satisfy the minimum size requirement, and (4) the summed functional group count equals the target value. The task completion rate is defined as the fraction of instances satisfying all four criteria. We show performance of GPT-5.4 on **REL**-C4 in Fig. A.3.

## B. Extended Background

### B.1. Relational Reasoning for Algebra

Algebra is a prototypical domain for relational reasoning because its primitives are defined by relations, most centrally equality and equivalence, and progress is made by applying transformations that preserve those relations. In typical mathematics tasks (solving equations, simplifying expressions, proving identities), a solver must maintain multiple constraints at once (e.g., which quantities are bound together, which substitutions are valid, which invariants are preserved) while manipulating symbolic structure. This emphasis on structure-preserving transformation makes algebra a natural setting for analyzing difficulty through the number of simultaneously active "slots" or variables that must be integrated in a single reasoning step, as formalized by relational complexity theory.

Raven's Progressive Matrices (John & Raven, 2003; Burke, 1972; Mills et al., 1993) can be viewed as an abstract completion problem in exactly this sense: the missing entry is determined by a rule that relates other entries, and solving requires inducing and composing those relations across the matrix. In **REL**-A, we use an algebraic reframing of RPMs (Camposampiero

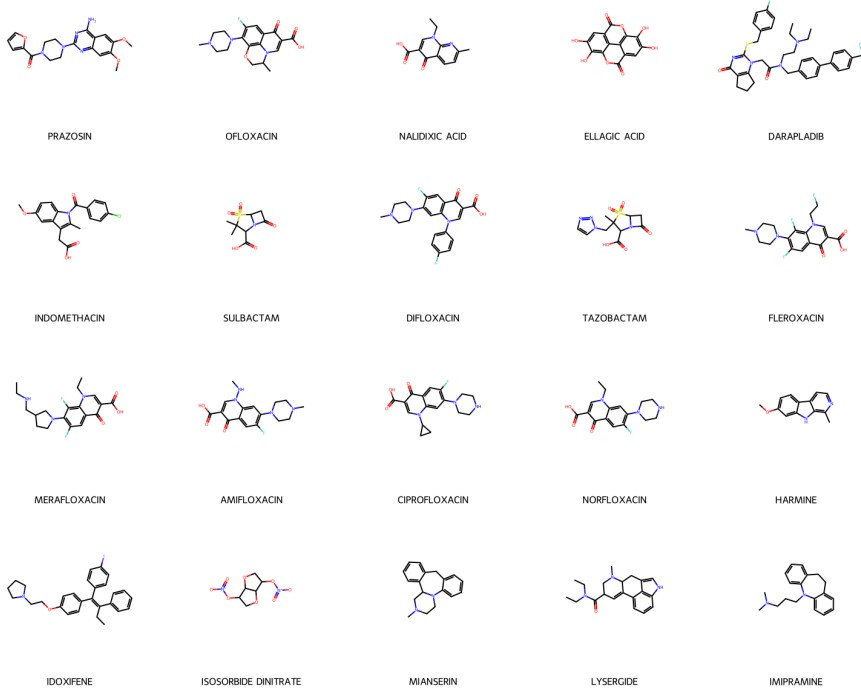

*Figure A.2.* Sample of 20 molecules used in **REL**-C2.

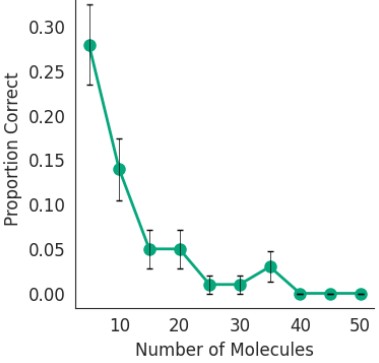

*Figure A.3.* Proportion correct as the task completion rate on **REL**-C4 stratified by RC which is given by $N_{\text{molecules}}$. Performance of GPT-5.4 is shown.

*Table A.5.* **REL**-C3. Number of missing isomers as a function of the number of molecules.

| $N_{\text{molecules}}$ | 5 | 10 | 15 | 20 | 25 | 30 | 35 | 40 | 45 | 50 |
|---|---|---|---|---|---|---|---|---|---|---|
| # Questions | 100 | 100 | 100 | 100 | 100 | 100 | 100 | 100 | 100 | 100 |
| # Missing Isomers | 36.41 | 32.92 | 29.93 | 30.39 | 24.13 | 27.28 | 24.17 | 33.24 | 28.16 | 26.44 |
| Avg. atoms | 5.91 | 5.83 | 5.84 | 6.22 | 6.23 | 6.19 | 6.42 | 6.18 | 6.22 | 6.31 |
| Avg. DBE | 4.56 | 4.69 | 4.73 | 4.74 | 4.59 | 4.78 | 4.66 | 5.39 | 5.34 | 5.40 |

*Table A.6.* Performance on **REL**-C3 across increasing RC which is determined by $N_{\text{observed}}$. Metrics reported are mean (s.e.) across the questions. For GPT 5.4 we provide the model access to RDKit.

| Model | Count | $N_{\text{observed}}$ | Recall | Precision | F1 |
|---|---|---|---|---|---|
| Claude Opus 4.5 | 300 | $< 20$ | 0.341 (0.013) | 0.569 (0.016) | 0.386 (0.011) |
| | 400 | 20–40 | 0.313 (0.011) | 0.390 (0.013) | 0.299 (0.008) |
| | 300 | $\geq 40$ | 0.253 (0.009) | 0.398 (0.015) | 0.289 (0.009) |
| Gemini 3 Pro | 300 | $< 20$ | 0.417 (0.014) | 0.472 (0.015) | 0.403 (0.012) |
| | 400 | 20–40 | 0.362 (0.012) | 0.275 (0.011) | 0.260 (0.009) |
| | 300 | $\geq 40$ | 0.259 (0.010) | 0.256 (0.012) | 0.234 (0.010) |
| GPT 5.2 | 300 | $< 20$ | 0.142 (0.011) | 0.410 (0.019) | 0.173 (0.010) |
| | 400 | 20–40 | 0.140 (0.008) | 0.198 (0.010) | 0.130 (0.006) |
| | 300 | $\geq 40$ | 0.102 (0.006) | 0.186 (0.011) | 0.124 (0.007) |
| GPT-5.4 + tools | 300 | $< 20$ | 0.109 (0.009) | 0.638 (0.021) | 0.160 (0.009) |
| | 400 | 20–40 | 0.094 (0.006) | 0.428 (0.018) | 0.130 (0.007) |
| | 300 | $\geq 40$ | 0.079 (0.005) | 0.470 (0.020) | 0.125 (0.006) |

et al., 2025b; Hersche et al., 2025) to control difficulty directly by controlling dependency: increasing the number of entries the missing value increases the arity of the governing relation and thus the relational complexity. Extending from matrices to Raven's Progressive Tensors (RPTs) further enlarges this design space, enabling local neighborhood rules (neighbor sums) whose relational bottleneck scales with the size of the dependency set.

### B.2. Relational Reasoning in Biology

In biology, many inferences require relational comparisons across organisms against an evolutionary backdrop. A canonical example is convergent evolution where the same mutation (or short sequence motif) arises independently in distinct lineages, which can indicate functional constraint or shared selection pressures (Stern, 2013; Storz, 2016) (e.g., viral adaptation (Markov et al., 2023; Bouhaddou et al., 2023), recurrent cancer tumor drivers (Gerlinger et al., 2012; McGranahan & Swanton, 2017)). In phylogenetics, such repeated, independent changes are referred to as homoplasy. Detecting homoplasy requires two inputs: (i) a shared motif in the multiple sequence alignment (MSA), and (ii) a phylogenetic tree establishing that the shared state is not explained by a recent common ancestor. Operationally, a motif shared by a subset of taxa is homoplasic if those taxa are evolutionarily separated on the tree (e.g., occupy different clades) such that shared ancestry alone cannot explain the pattern (Crispell et al., 2019; Wake, 1991).

### B.3. Relational Reasoning in Chemistry

Many benchmarks evaluate LLMs on their ability to reason for questions that require chemistry domain knowledge, including ChemLLMBench (Guo et al., 2023), ChemEval (Huang et al., 2024), ChemBench (Mirza et al., 2025), and ScholarChemQA (Chen et al., 2025). Several benchmarks also evaluate the ability of models for SMILES comprehension. For example, Mol-Instructions (Fang et al., 2023), CleanMol (Jang et al., 2025), and ChemIQ (Runcie et al., 2025) evaluate the model ability for graph-level molecular comprehension. ChemCoTBench (Li et al., 2025) involves granular tasks with molecular editing of SMILES for property optimization and reaction prediction. FGBench (Liu et al., 2025b) focuses on functional group-level reasoning for molecular property prediction. Previous benchmarks focus primarily on individual molecules or, at most, reactant-product relationships in chemical reactions, limiting their ability to evaluate whether LLMs can reason across sets of molecules and infer higher-order relations.

Relational reasoning with the understanding of shared structure across molecules is a fundamental aspect of chemistry. Bemis & Murcko (1996) introduced a formal definition of molecular scaffolds to organize and compare large collections

*Table A.7.* Performance on open source models for 500 questions from REL-A2 and 500 questions from REL-A3.

| Model | Rule Type (RC) | 3x3 | 9x9 | 15x15 | 30x30 |
|---|---|---|---|---|---|
| Llama 3.1-70b | Progression (2) | 0.912 | 0.864 | 0.856 | 0.816 |
| Llama 3.1-70b | Permutation (n) | 0.648 | 0.552 | 0.296 | 0.144 |
| Qwen3-4B | Progression (2) | 0.616 | 0.568 | 0.528 | 0.504 |
| Qwen3-4B | Permutation (n) | 0.552 | 0.216 | 0.184 | 0.104 |

of drug-like molecules, influencing how chemical space is structured and explored in drug discovery. This scaffold-based view underpins scaffold hopping, a standard strategy for exploring new druggable regions of chemical space (Acharya et al., 2024). More broadly, identifying chemical relationships at scale is central to library design (Hajduk et al., 2011) and the analysis of structure–activity relationships (Guha, 2013), both of which aim to enable more systematic exploration of vast chemical spaces.

## C. Extended Related Work: Relational Complexity in Other Benchmarks

In this section, we provide a brief overview of how relational complexity as introduced in this paper appears in other existing benchmarks. We believe that much insight could be gained from re-examining LLM results on these benchmarks through the lens of RC and that more difficult future benchmarks in these areas will naturally incorporate higher RC settings.

**Reasoning on graphs.** Existing benchmarks such as VisionGraph (Li et al., 2024b) that have LLMs execute graph algorithms such as shortest path, bi- or multi-partite matching, or message passing adopt the size of the graph as the primary notion of difficulty. RC enters these in problems in the form of the (average) neighborhood density in shortest path search, the number of independent sets in multipartite matching, and the size of the receptive field in message passing. We believe that these are much more natural notions of instance difficulty than the number of nodes in a graph.

**Visual reasoning and autonomous driving.** Benchmarks such as MMT-Bench (Ying et al., 2024) or MME-RealWorld (Zhang et al.) contain tasks such as scene-graph generation, multi-object reasoning, or multi-image comparison. In each of these settings, relational complexity naturally enters as the number of entities involved. Perhaps most importantly, in autonomous driving (Zhang et al.), models are required to reason over images with multiple vehicles or pedestrians and must decide on the right course of action. Existing benchmarks only test this when the number of entities is low. The much more challenging high RC setting, i.e. potentially chaotic traffic situations with dozens of participants involved, has received much less coverage.

**Medical reasoning.** Medical benchmarks are often treated as if difficulty were driven mainly by note length, terminology, or disease rarity (Dumas et al., 2014a). From the RC perspective, the more natural driver is the number of interacting clinical entities and constraints—symptoms, comorbidities, medications, lab trends, imaging findings, and time. Low-RC questions allow near-local pattern matching, whereas high-RC cases require integrating multiple signals, resolving contradictions, and tracking relations over time (e.g., drug-drug interactions or treatment response). We expect that re-examining medical benchmark results through RC will better explain common failure modes (single-cause shortcuts, missed interactions, poor temporal tracking) and that more realistic evaluations will increasingly emphasize high-RC, longitudinal patient trajectories.

## D. Additional Experiments

### D.1. Open-Weights Models

We present additional results with Qwen3-4B and Llama3.1-70B on a subset of the **REL**-A tasks in Table A.7. Both models support our main claim that relational complexity tracks performance better than input size, though Llama is stronger than Qwen.

### D.2. In-Context Learning

In Fig A.4 we how one-shot in-context learning changes the task completion rate across **REL**-C1,C2,C3.

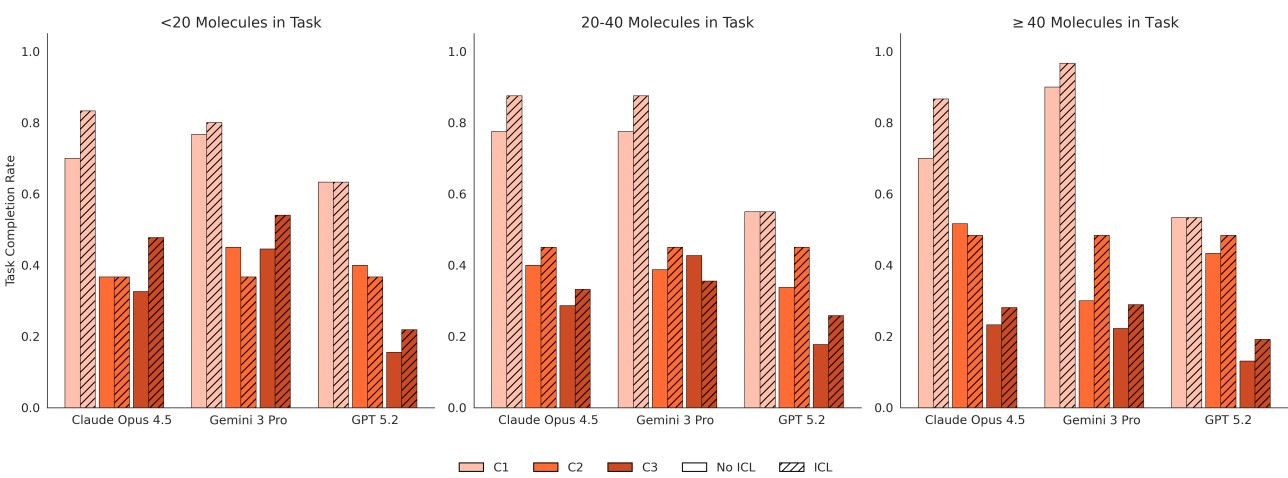

*Figure A.4.* Models are evaluated with one-shot in-context learning on 10% of questions from **REL**-C1,C2,C3.

### D.3. Test-Time Compute

In Table A.8, we investigate whether additional test-time compute can improve performance on difficult **REL**-A tasks.

*Table A.8.* Accuracy by number of thinking tokens on small examples of **REL**-A4 and **REL**-A5. Gains are small, but consistent.

| Digits | GPT REL-A5 ($3 \times 3 \times 3$) | Claude REL-A5 ($3 \times 3 \times 3$) | GPT REL-A4 ($9 \times 9$) | Claude REL-A4 ($9 \times 9$) |
|---|---|---|---|---|
| 4096 Tokens | 0.648 | 0.616 | 0.152 | 0.152 |
| 8192 Tokens | 0.664 | 0.624 | 0.168 | 0.144 |
| 16384 Tokens | 0.680 | 0.648 | 0.176 | 0.168 |

In Fig A.5 we show how test-time compute changes the task completion rate across **REL**-C1,C2,C3.

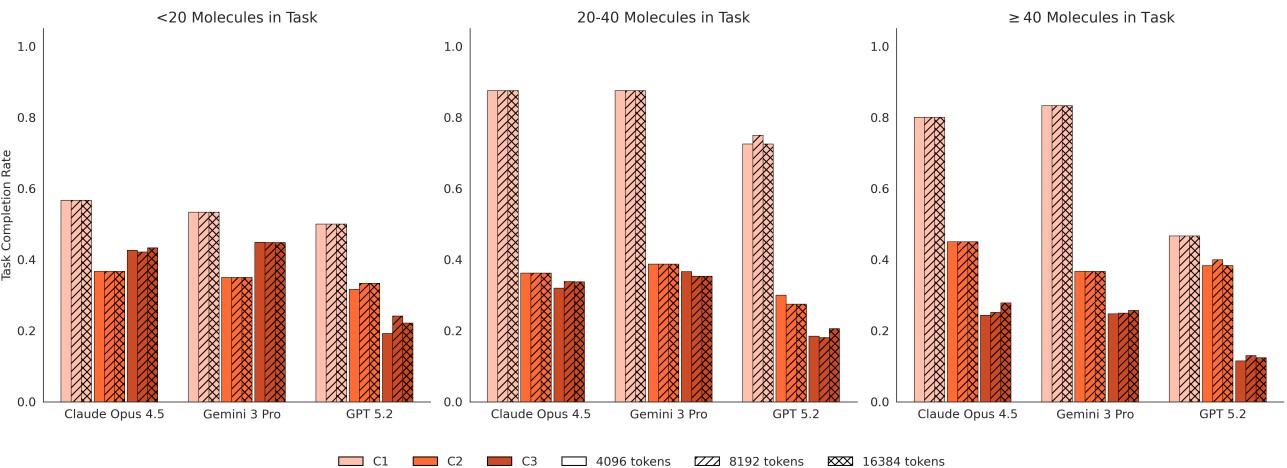

*Figure A.5.* Models are evaluated with different levels of test-time compute on 10% of questions from **REL**-C1,C2,C3.

### D.4. Operand Complexity

**Operand Complexity.** We vary the OC for **REL**-A3 and REL-A4 tasks by increasing the number of digits per entry in the RPM from 5 to 10 and then 20. As Table A.9 shows, moving from 5 to 10 digits results in only a small drop between 1 and 6% accuracy. Increasing this to 20 digits leads to catastrophic accuracy drops of as much as 50%, which seems to hint at the well-known difficulty that LLMs have when representing numbers (Zhou et al., 2025).

*Table A.9.* Accuracy by number of digits per entry on small examples of **REL**-A3 and **REL**-A4. Only for extremely large input values does accuracy plummet.

| Digits | GPT REL-A3 ($9 \times 9$) | Claude REL-A3 ($9 \times 9$) | GPT REL-A4 ($3 \times 3$) | Claude REL-A4 ($3 \times 3$) |
|---|---|---|---|---|
| 5 Digits | 0.912 | 0.752 | 0.248 | 0.208 |
| 10 Digits | 0.872 | 0.688 | 0.184 | 0.192 |
| 20 Digits | 0.360 | 0.336 | 0.136 | 0.112 |

We perform an analogous manipulation of OC for tasks **REL**-B1 by varying the motif ratio. As shown in Table A.10, decreasing the motif ratio from 10–12.5% to 0–5% causes average model accuracy to collapse by 92%, approaching zero across all models. Accuracy is highest at intermediate motif ratios and degrades sharply when the motif occupies only a small fraction of the sequence, indicating that models struggle to identify relevant structure when most of the input is irrelevant. This mirrors known difficulties of LLMs on needle-in-a-haystack settings (Bianchi et al., 2025). Together, these results demonstrate that OC, independent of RC, exerts a strong and systematic influence on model accuracy.

*Table A.10.* Accuracy by motif ratio (motif size / sequence length). Performance collapses as operand complexity increases in small motif ratios.

| Motif Ratio | Gemini | GPT | Claude |
|---|---|---|---|
| 0–5% | 2.6% | 0.8% | 0.0% |
| 5–10% | 9.2% | 10.2% | 1.8% |
| 10–12.5% | 13.5% | 18.6% | 2.1% |
| 12.5–15% | 22.6% | 29.0% | 12.9% |
| 15–17.5% | 19.6% | 17.6% | 17.6% |
| 17.5–20% | 34.2% | 23.7% | 28.9% |
| 20–22.5% | 18.2% | 21.9% | 10.9% |
| 22.5–25% | 28.6% | 20.0% | 22.9% |
| 25–30% | 31.2% | 28.0% | 16.1% |

### D.5. Structured Prompting

A key question is whether the observed performance degradation at high relational complexity (RC) is due to a failure of procedural reasoning, and whether explicit decomposition into sequential steps could mitigate this limitation.

To test this directly, we introduced a structured prompting strategy that enforces a step-by-step decomposition of the **REL**-B1 task (Prompt D.5). Specifically, the model is required to: (1) identify local motif-sharing groups across alignment columns, (2) filter for groups that consistently co-occur across independent positions, (3) evaluate phylogenetic distances to distinguish homoplasy from shared ancestry, and (4) produce a final answer based on these intermediate results. This decomposition reflects the minimal sequential procedure required to solve the task and is designed to reduce the need for simultaneous reasoning over all taxa .

> **Structured Prompt**
>
> **Instruction:** You MUST follow these steps in order. Show your work for each step.
> **Step 1 — Motif scanning:** Scan through the alignment columns in blocks. For each block of 10 consecutive columns, identify which taxa share identical nucleotide patterns (motifs) at those positions. List the groups of taxa that share motifs.
> **Step 2 — Consistency check:** Among the groups you found in Step 1, identify which groups of taxa CONSISTENTLY share motifs across MANY independent columns (not just a few). A group is significant only if the same taxa repeatedly co-occur with matching nucleotides across many columns.
> **Step 3 — Tree distance check:** For each consistent group from Step 2, use the provided phylogenetic tree to determine whether the taxa in that group are closely related (share a recent common ancestor) or distantly related (separated by long branches). Only groups of DISTANTLY related taxa constitute homoplasy.
> **Step 4 — Final answer:** Based on Steps 1-3, provide your final answer.

We evaluated this structured prompting on newly generated **REL**-B1 instances with varying numbers of homoplastic taxa (RC $\in \{4, 10, 20, 25\}$) (Prompt D.5).

Our results (Table A.11) indicate that while structured reasoning can partially assist when the number of interacting entities is

*Table A.11.* Accuracy (%) on **REL**-B1 under increasing relational complexity (RC). Structured prompting improves performance at low RC but yields diminishing gains as RC increases.

| | Relational Complexity (RC) | | | |
|---|---|---|---|---|
| Method | 4 | 10 | 20 | 25 |
| Baseline | 45% | 15% | 5% | 5% |
| Structured | 65% | 25% | 10% | 5% |
| $\Delta$ | +20 | +10 | +5 | 0 |

small, it does not eliminate the performance degradation at high RC. Critically, even when the task is explicitly decomposed into sequential subtasks, the model must still maintain and integrate information about many taxa simultaneously to ensure global consistency. This suggests that the primary bottleneck is not the absence of an appropriate reasoning procedure, but rather the difficulty of maintaining multiple bindings in parallel within the model's internal representations.

Overall, this experiment provides evidence that relational complexity reflects an intrinsic limitation in current models' ability to handle many-way interactions, rather than a failure that can be resolved through prompt-level decomposition alone.

### D.6. Best-of-N and Majority-Vote Aggregation on REL-B1

We also performed an exploratory analysis of alternative test-time compute strategies on **REL**-B1. Specifically, we randomly sampled 15 questions at each of four levels of biological relational complexity ($n_{ht} \in 4,10,20,25$) and ran GPT-5.2 five times per question. From these runs, we evaluated two aggregation strategies: best-of-$NN$ (BoN), where we report the best result across the five samples, and majority vote (MajVote), where a taxon is included only if it is predicted in the majority of runs.

We found that these strategies did not substantially recover performance at higher RC. Under BoN, accuracy improved from 46.7% to 60.0% at $n_{ht} = 4$, but remained unchanged at $n_{ht} = 10$ and $n_{ht} = 20$, and increased only slightly from 0% to 6.7% at $n_{ht} = 25$. Under MajVote, performance generally decreased, from 46.7% to 40.0% at $n_{ht} = 4$, from 20.0% to 6.7% at $n_{ht} = 10$, remained at 6.7% at $n_{ht} = 20$, and remained at 0% at $n_{ht} = 25$. This analysis suggests that simple self-consistency-style test-time compute does not materially alleviate the degradation observed at high RC.

### D.7. Qualitative Analysis of Failure Cases

We conducted a qualitative analysis of model outputs on **REL**-B1 to better understand typical failure modes at high relational complexity. We observed that the models did not simply collapse into random guessing; instead, each exhibited a distinct and recurring error pattern. Claude often identified relevant taxa or clades early in its reasoning, but later reversed earlier conclusions and ultimately concluded that no significant homoplasy was present. GPT-5.2 showed a positional bias, disproportionately selecting higher-index taxa even when the evidence did not support those choices, and its reasoning sometimes focused on irrelevant properties such as GC-rich motifs or unrelated clade structure. Gemini did not exhibit the same positional bias, but showed systematic under-counting at higher RC: it frequently identified part of the convergent group while omitting additional homoplastic taxa. In the **REL**-A3 task we observed similar behavior in Claude where the model discusses several possible rules that could generate the RPM but then goes into circles instead of digging into one, thereby exhausting the token budget. If the model decides on a suspected rule, it is usually the right one (permutation), and the model is then almost always able to deduce the correct missing value.

## E. Example Prompts

Below we provide example prompts of each of the tasks in **REL**.

### E.1. REL-Algebraic

**REL**-A1

```
Only return the missing number!
row 1:  633, 633, 633; row 2:  354, 354, 354; row 3:  761, 761,
Answer set:
```

```
Answer #0:  769
Answer #1:  781
Answer #2:  789
```
**Answer #3:  761**
```
Answer #4:  780
Answer #5:  752
Answer #6:  712
Answer #7:  743
```

## REL-A4

```
Only return the missing number!
row 1:  723, 38, 761; row 2:  152, 204, 356; row 3:  233, 279,
Answer set:
Answer #0:  476
Answer #1:  502
Answer #2:  334
Answer #3:  255
```
**Answer #4:  512**
```
Answer #5:  417
Answer #6:  687
Answer #7:  780
```

**REL**-A5

```
Complete the Raven's progressive tensor:
Only return the missing number!
Slice l=0:
row 1:  3, 6, 5
row 2:  4, 8, 9
row 3:  1, 7, 9
Slice l=1:
row 1:  13, 24, 29
row 2:  17, 25, 31
row 3:  17, 22, ?
Slice l=2:
row 1:  76, 84, 114
row 2:  78, 88, 115
row 3:  77, 88, 112
Answer set:
Answer #0:  45
Answer #1:  7
Answer #2:  52
Answer #3:  32
Answer #4:  24
Answer #5:  30
Answer #6:  16
Answer #7:  63
```

**REL**-A6

```
Complete the Raven's progressive tensor:
Only return the missing number!
Slice l=0:
row 1:  3, 6, 5
row 2:  4, 8, 9
row 3:  1, 7, 9
Slice l=1:
row 1:  8, 7, 10
row 2:  3, 1, 2
row 3:  2, 9, ?
Slice l=2:
row 1:  8, 2, 3
row 2:  5, 0, 3
row 3:  9, 6, 10
```

```
Answer set:

Answer #0:  0

Answer #1:  1

Answer #2:  8

Answer #3:  3

Answer #4:  6

Answer #5:  5

Answer #6:  10

Answer #7:  9
```

## E.2. REL-Biology

### REL-B1

```
Homoplasy refers to structured convergence:

pairs or groups of distantly related taxa that repeatedly share the same
nucleotide motifs

across many independent alignment columns more often than expected, while
other taxa

with similar overall sequences do not share those nucleotide motifs as
consistently.

Your job is to examine the entire alignment and provided tree and decide
whether such structured

homoplasy is likely to be present and which taxa are involved.

Alignment (FASTA; positions indexed from 1):

>6

GAGATAATCATTCGGGAGTCAATTCCAAAATCCGTTCGGGATGAATTGTCTATCTGCCCCGCTTCGTGAGTACCGCTAACTCCTCG

... (rest of sequences) ...

Tree (Newick):

(((6:0.9078,(3:0.8576,46:0.6305):0.5442):0.4086,(((((12:0.6359,(5:0.3115,16:0.3136):

... (rest of tree) ...

Return your answer as:  Yes/No and if Yes, list the taxa involved.
```

### REL-B2

```
A protein has been measured with the following mutations at 2 positions.

Below are the measured fitness values for all 4 combinations:

Genotype Fitness

wild-type 2.401243

A54A 1.751533

V39C 1.249425

V39C + A54A 0.566714

Which of the following is the best explanation of the full table?

A. V39C and A54A modify each other's effects | the double mutation fitness
is HIGHER than predicted by adding each mutation's individual effect.
```

B. V39C and A54A modify each other's effects | the double mutation fitness is LOWER than predicted by adding each mutation's individual effect.

C. V39C and A54A act independently | the double mutation fitness is well predicted by adding each mutation's individual effect.

Answer with just the letter.

## E.3. REL-Chemistry

### REL-C1

Is this list of molecules a set of *constitutional isomers* (same molecular formula, different connectivity)?
SMILES:
1.  ClC1C(Cl)C1Cl
2.  CC(Cl)=C(Cl)Cl
3.  C=CC(Cl)(Cl)Cl
4.  ClC=CC(Cl)Cl
5.  ClCC=C(Cl)Cl

Return exactly one of:
<Yes>
or
<No>
No explanation.

### REL-C2

Given the following list of SMILES, what is the largest *connected* common chemical motif (maximum common substructure) present in every molecule?
Rules:
- The motif must be a single connected fragment.
- Do NOT tautomerize molecules.
- Ignore stereochemistry unless it is explicitly encoded and required.

SMILES:
1.  COc1ccc2c(c1)N(CC(C)CN(C)C)c1ccccc1S2
2.  CC(CN(C)C)CN1c2ccccc2Sc2ccccc21
3.  CCc1ccc2c(c1)N(CC(C)CN(C)C)c1ccccc1S2
4.  CSc1ccc2c(c1)N(CC(C)CN(C)C)c1ccccc1S2
5.  CC(CN(C)C)CN1c2ccccc2Sc2ccc(C#N)cc21

Return your final answer as a single SMILES wrapped exactly like:
<smiles>YOUR_SMILES_HERE</smiles>
No explanation.

### REL-C3

Given the following list of constitutional isomers, complete the set by identifying the missing constitutional isomers.

Given SMILES:
1.  FCCC1CC1
2.  C=C(F)C(C)C
3.  CCC1CC1F
4.  C=CCCF

5.  `C=CCC(C)F`

Return the missing molecules as SMILES, one per line, each wrapped exactly like:
`<smiles>YOUR_SMILES_HERE</smiles>`
No explanation.

## REL-C4

Given the following 5 molecules, identify one continuous motif from **each** molecule.

**Task**

1. From each of the 5 molecules below, extract one continuous motif (substructure).
2. Ensure the total count of total_carboxylic_acids across all motifs equals 1.

**Constraints**

- Each motif must be a valid SMILES string (complete and parseable by RDKit).
- Each motif must be a substructure that actually exists in its parent molecule.
- Each motif must contain at least 6 heavy atoms (non-hydrogen).
- The sum of total_carboxylic_acids across all selected motifs must equal 1.

**Critical validation rules**

- SMILES must be complete; do not truncate or abbreviate.
- Rings must be closed: every ring opening digit (1--9) must have a matching closing digit.
- Wrong: `CC12CCC(=O)C=C1` (ring 2 never closes) --- invalid SMILES.
- Right: `CC12CCC(=O)C=C1CC2` (both rings 1 and 2 close properly).
- Each motif must be a continuous fragment that exists exactly as written in its parent molecule.
- When extracting from complex fused rings, use simpler motifs if needed.
- Count total_carboxylic_acids carefully.
- Verify that the total sum equals 1 before submitting.

**Molecules**

1. `CCN(CC)C(C)=NN=Cc1c2c(O)c3c(O)c(C)c4c(c3c1O)C(=O)C(C(OC=CC(OC)C(C)C(OC(C)=O)C(C)C(O)C(C)C(O)C(C)C=CC=C(C)C(=O)N2)O4`
2. `CCC1OC(=O)C(C)C(=O)C(C)C(OC2OC(C)CC(N(C)C)C2O)C2(C)CC(C)C(=NC(C)=O)C(C)C(OCC(=NOCc3ccc(-n4cccn4)nc3)CO2)C1(C)O`
3. `CCC1OC(=O)C(C)C(=O)C(C)C(OC2OC(C)CC(N(C)C)C2O)C(C)(OC)CC(C)C(=O)C(C)C2C(C(N)=NOC(C)c3nnc(-c4ccccn4)s3)C(=O)OC12C`
4. `CCC12CN3CCc4c([nH]c5ccccc45)C(C(=O)OC)(c4cc5c(cc4OC)N(C=O)C4C(O)(C(=O)OC)C(OC(C)=O)C6(CC)C=CCN7CCC54C76)CC(C3)C1O2`
5. `CCOC(=O)CCC(=O)OC1C(OC2C(C)C(OC3CC(C)(OC)C(O)C(C)O3)C(C)C(=O)OC(CC)C(C)(O)C(O)C(C)C(=O)C(C)CC2(C)O)OC(C)CC1N(C)C`

**Step-by-step approach**

1. For each molecule, identify candidate motifs with at least 6 heavy atoms.
2. Count total_carboxylic_acids in each candidate motif.
3. Select one motif from each molecule such that the total sum equals 1.
4. Some motifs may contain 0 total_carboxylic_acids; this is allowed.
5. Extract the exact substructure from the parent molecule and copy it precisely.
6. Ensure each SMILES is complete, with all rings properly closed (e.g., `c1ccccc1`).
7. Final check: each motif exists in its parent molecule and the total sum equals 1.

**Functional group examples (for reference)**

- Ketone:  C(=O)C or CC(=O)CC
- Carboxylic acid:  C(=O)O or CC(=O)O
- Ester:  C(=O)OC or CC(=O)OC
- Aldehyde:  C(=O) at chain end
- Primary amine:  CNH2 or CCN
- Alcohol:  CO (hydroxyl on an sp3 carbon)
- Aromatic ring:  c1ccccc1 (benzene)

**Output format** (indices are 0-indexed and must include all molecules)

```
<indices>0,1,2</indices>
<motif_0>CCCCCC</motif_0>
<motif_1>c1ccccc1</motif_1>
<motif_2>CC(=O)O</motif_2>
```

**Format rules**

- List all molecule indices in the <indices> tag (0 through 4), comma-separated.
- For each index, provide a complete motif SMILES in the corresponding <motif_N> tag.
- Do not use <smiles> tags; use <motif_N> where $N$ is the molecule index.
- SMILES must be complete (e.g., c1ccccc1, not c1ccc).

**Critical reminder** To obtain 1 total_carboxylic_acids:

- You must provide a motif for every molecule (all 5 molecules).
- Some motifs may have 0 total_carboxylic_acids; balance is key.
- Adjust motif selections so that the total equals 1.

**Before submitting, verify**

- Provided a motif for all 5 molecules (indices 0 through 4).
- Each SMILES is complete and valid, with all rings closed.
- Each motif exists in its parent molecule.
- Counted total_carboxylic_acids in each motif.
- The sum of total_carboxylic_acids is exactly 1.

Provide only the formatted answer above.  No explanation.

