# OpenReview forum: "Evaluating Relational Reasoning in LLMs with REL"
_ICML.cc/2026/Conference — ICML 2026 regular_

### Official Review · Reviewer_5maR · 2026-02-24

**Soundness:** 3
**Presentation:** 3
**Significance:** 3
**Originality:** 3
**Overall Recommendation:** 4
**Confidence:** 3

**Summary:**

This paper explores the limitations of Large Language Models (LLMs) in relational reasoning through the lens of Relational Complexity (RC). The authors define RC as the minimum number of independent entities that must be simultaneously bound to apply a relation, distinguishing it from superficial difficulty markers like input size. They introduce REL, a generative benchmark spanning algebraic (RPMs/Tensors), biological (homoplasy detection), and chemical (isomers and substructures) domains. Evaluating frontier models such as GPT-5.2 and Claude Opus 4.5, the study reveals a consistent, monotonic degradation in performance as RC increases, even when the number of entities remains fixed. Overall, this manuscript propose that current LLM architectures face a fundamental bottleneck in higher-arity relational binding, a failure mode that persists despite increased test-time compute or in-context learning.

**Compliance With Llm Reviewing Policy:**

Affirmed.

**Final Justification:**

My concerns have been adequately addressed. I will maintain my positive score.

**Key Questions For Authors:**

Refer to the above part

**Limitations:**

yes

**Strengths And Weaknesses:**

Strength
1. The authors investigate a highly interesting problem with significant potential for extension into additional scientific and logic-based domains.
2. An important strength is the integration of Relational Complexity (RC) from cognitive science with machine learning evaluation methodology. This approach probes the essence of reasoning difficulty more effectively than simply increasing input length or entity counts.
3. The REL benchmark is not limited to a single domain; spanning from abstract tensor operations to specific evolutionary biology and molecular chemistry, the consistency of experimental results across these fields greatly enhances the persuasiveness of the study's conclusions.
4. The analysis is remarkably comprehensive; beyond reporting raw accuracy, the authors conduct exhaustive ablation experiments on Operand Complexity (OC), Chain-of-Thought (CoT) token length, and In-Context Learning (ICL).

Weakness
1. REL is currently built primarily on synthetic data. While this is necessary for strict variable control, the impact of the paper would be further elevated if the authors explored how the RC bottleneck manifests in real-world scientific literature processing, such as the analysis of complex metabolic pathways.
2. It remains unclear whether models are merely engaging in random guessing under high RC conditions or if they fall into specific logical loops. Including a qualitative analysis of typical failure cases would be highly illuminating.
3. The evaluation is restricted to closed-source API models. It would be valuable to know if these findings—specifically the sharp performance collapse at certain RC thresholds—are also observable in smaller-scale open-weight models.

---

> ### Author Rebuttal · Authors · 2026-03-30
>
> > REL is currently built primarily on synthetic data. While this is necessary for strict variable control, the impact of the paper would be further elevated if the authors explored how the RC bottleneck manifests in real-world scientific literature processing, such as the analysis of complex metabolic pathways.
>
> We thank the reviewer for this suggestion and have included a new biology task, REL-B2 that is centered around uncovering epistatic structure. Specifically, we use five experimentally measured protein fitness landscapes (GFP[1], HA[2], HA2[2], GB1[3], TrpB[4]), spanning antibody binding and enzyme activity.
> Each task instance is constructed by selecting k focal mutations from a real landscape, fixing the remaining positions as a background, and retaining only cases where all 2^k combinations are experimentally observed. This defines a fully measured local genotype–fitness table. We then compute standard Walsh–Hadamard epistatic coefficients to assign a coarse interaction-structure label (e.g., approximately independent, dominant pair, higher-order modulation). The model does not observe these coefficients; it must infer the correct structure solely from the raw fitness table.
> This preserves the controlled notion of relational complexity (RC = k) while requiring reasoning over real sequence–function data. Empirically, we observe the same qualitative behavior as in the synthetic setting: models perform strongly at low complexity (e.g., ~76–83% accuracy at REL-2 across datasets) but degrade sharply toward chance as RC increases. This demonstrates that the RC bottleneck is not an artifact of synthetic data, but persists in real biological landscapes. We have added this extra experiment and results to the paper. [1] https://pmc.ncbi.nlm.nih.gov/articles/PMC6746860/#Sec2 [2] https://pmc.ncbi.nlm.nih.gov/articles/PMC8476123/#s7 [3] https://elifesciences.org/articles/16965 [4] https://pmc.ncbi.nlm.nih.gov/articles/PMC11317637/
>
> > It remains unclear whether models are merely engaging in random guessing under high RC conditions or if they fall into specific logical loops. Including a qualitative analysis of typical failure cases would be highly illuminating.
>
> We additionally conducted a qualitative analysis of model outputs on REL-B1 to better understand typical failure modes at high relational complexity. We observed that the models did not simply collapse into random guessing; instead, each exhibited a distinct and recurring error pattern. Claude often identified relevant taxa or clades early in its reasoning, but later reversed earlier conclusions and ultimately concluded that no significant homoplasy was present. GPT-5.2 showed a positional bias, disproportionately selecting higher-index taxa even when the evidence did not support those choices, and its reasoning sometimes focused on irrelevant properties such as GC-rich motifs or unrelated clade structure. Gemini did not exhibit the same positional bias, but showed systematic under-counting at higher RC: it frequently identified part of the convergent group while omitting additional homoplastic taxa. We observe a similar pattern in REL-A3: Claude frequently considers multiple candidate rules that could explain the data, but fails to commit to one, instead looping over alternatives and exhausting the token budget. These observations suggest that performance degradation at high RC is not driven purely by random error, but by structured failure modes in how different models track, maintain, and integrate multiple interacting entities.
>
> >The evaluation is restricted to closed-source API models. It would be valuable to know if these findings—specifically the sharp performance collapse at certain RC thresholds—are also observable in smaller-scale open-weight models.
>
> We agree with the reviewer that investigating smaller open-weight models would be a valuable extension to the current evaluation. Below, we present additional results with Qwen3-4B and Llama3.1-70B (explicitly requested by reviewer e2ke) on a subset of the REL-A tasks. The numbers from both models support the main message of the original submission, i.e. relational complexity tracks performance better than other metrics such as input size, although Llama is a clearly stronger model than Qwen here. We will test both models on all tasks for the final manuscript.
>
> - Llama3.1-70B:
>
> | Rule Type (RC)  |   3x3 |   9x9 | 15x15 | 30x30 |
> | --------------- | ----: | ----: | ----: | ----: |
> | Progression (2) | 0.912 | 0.864 | 0.856 | 0.816 |
> | Permutation (n) | 0.648 | 0.552 | 0.296 | 0.144 |
>
> - Qwen3-4B:
>
> | Rule Type (RC)  |   3x3 |   9x9 | 15x15 | 30x30 |
> | --------------- | ----: | ----: | ----: | ----: |
> | Progression (2) | 0.616 | 0.568 | 0.528 | 0.504 |
> | Permutation (n) | 0.552 | 0.216 | 0.184 | 0.104 |

---

> > ### Author Rebuttal · Reviewer_5maR · 2026-04-03
> >
> > Thanks to the authors for their careful work. They have addressed my concerns, and I acknowledge the quality of the paper. I will maintain my positive score.

---

### Official Review · Reviewer_e2ke · 2026-03-04

**Soundness:** 3
**Presentation:** 3
**Significance:** 4
**Originality:** 3
**Overall Recommendation:** 5
**Confidence:** 4

**Summary:**

REL introduces Relational Complexity (RC)—the minimum number of independent information sources that must be simultaneously bound during a single reasoning step—as a diagnostic axis for evaluating LLM reasoning. The benchmark spans three scientific domains: algebra (numerical Raven's Progressive Matrices and a new 3D extension called RPT), biology (homoplasy detection on phylogenetic trees), and chemistry (constitutional isomer analysis). Evaluation on Claude Opus 4.5, Gemini 3 Pro, and GPT-5.2 shows RC is the dominant predictor of performance degradation across all three domains. Increasing test-time compute (thinking tokens up to 16,384) yields only 2-3% improvement on high-RC tasks, and one-shot ICL cannot close the gap.

**Compliance With Llm Reviewing Policy:**

Affirmed.

**Final Justification:**

The paper makes a strong and original contribution by introducing Relational Complexity (RC) as a diagnostic axis for LLM reasoning, and by showing its effect consistently across three structurally distinct domains: algebra, biology, and chemistry. The strongest aspect of the submission is this cross-domain consistency. Although the underlying tasks differ substantially in form and mechanism, performance degrades monotonically as RC increases, and the reported regression analysis further supports RC as a stronger predictor of failure than competing factors. The benchmark design is also practically valuable, since all three components are generated procedurally, which improves scalability and reduces contamination concerns.

My main concerns from the original review were meaningfully addressed in the rebuttal. The authors clarified the distractor construction in REL-A and provided evidence that the nearest-distractor distance remains approximately stable across RC levels, which substantially reduces the concern that low-RC performance might be inflated by answer elimination effects. They also added open-weight model results on REL-A subsets, which strengthens the claim that the observed RC pattern is not limited to a small set of proprietary APIs. In addition, they reported precision and F1 for REL-C3, addressing the concern that recall alone might overstate performance through over-generation. Finally, they supplemented the test-time compute analysis with exploratory best-of-N and majority-vote experiments, and these results further support the conclusion that simple additional inference-time compute does not materially mitigate degradation on high-RC tasks.

Some limitations remain, particularly around how strongly RC should be compared across domains in an absolute sense, and the final version would still benefit from improved uncertainty reporting and broader open-model coverage. However, I view these as opportunities for refinement rather than flaws that undermine the core contribution. Overall, the paper is technically solid, conceptually interesting, and potentially influential for future reasoning evaluation work. In light of the rebuttal, I believe the key concerns have been adequately resolved, and I remain supportive of acceptance.

**Key Questions For Authors:**

1. What is the distribution of distractor values relative to correct answers in REL-A? Specifically, what is the mean absolute difference between the correct answer and the nearest distractor, and does this vary systematically with RC? If distractors are well-calibrated (i.e., numerically close to the correct answer), the elimination concern goes away. If not, the RC effect size in Figure 5 may be inflated for low-RC tasks.

2. Have you considered evaluating even one open-source model on a REL-A subset? The "RC is model-agnostic" claim currently rests on three closed-source models from three providers. A single open-source result (even partial) would substantially broaden the evidence base and improve reproducibility.

3. For REL-C3, what are the precision values alongside the reported recall? If precision is low (models generating many candidate SMILES), the recall-only metric may overstate actual isomer identification ability.

4. The test-time compute analysis uses extended thinking tokens. Have you tested other forms of test-time compute (e.g., best-of-N sampling, self-consistency) on high-RC tasks? If those also fail, the conclusion about RC as a fundamental bottleneck becomes stronger.

**Limitations:**

Partially. The paper discusses limitations in lines 428-434 (multiple-choice format, context-length constraints, synthetic tasks). But it omits: closed-source model reproducibility risk, distractor generation transparency, absence of any mitigation experiments, and the RC cross-domain comparability assumption. The Impact Statement (lines 440-451) appropriately discusses deployment implications.

**Strengths And Weaknesses:**

Strengths:

The cross-domain consistency is the paper's strongest evidence. Algebra, biology, and chemistry are designed with completely independent logic—RPM rules vs phylogenetic tree traversal vs molecular graph analysis—yet all three show monotonic performance decline with RC. In REL-B, the multivariate regression (Section 5.2, Figure 7) quantifies this: RC explains 24% of variance for Claude, 32% for Gemini, 44% for GPT, while the next strongest factor explains only 6-17%. Three independent domains showing the same pattern is much harder to dismiss than a single-domain result.

The test-time compute finding (Section 5.4, Table A.5) has value beyond this benchmark. Thinking tokens 4096→16384 gives +2-3% on REL-A high-RC tasks and +0.4% average on REL-C (lines 425-431). This is a concrete data point against the narrative that scaling inference compute solves reasoning bottlenecks. Though I'd note this only tests one form of test-time compute (extended thinking)—it doesn't rule out that other approaches like tree search or self-consistency might help more.

The generative framework is a practical contribution. All three sub-benchmarks are parameterized generators (Algorithm 1 for REL-A, Pyvolve for REL-B, Surge for REL-C), so new instances can be generated indefinitely. In an era of increasing data contamination concerns, this matters.

RC as a task-agnostic difficulty axis is genuinely new in LLM evaluation. I-RAVEN-X (Camposampiero et al., 2025b) also uses numerical RPMs but focuses on generalization and robustness (longer rules, wider attribute ranges, perceptual noise). REL's contribution is different: it operationalizes RC from cognitive science as a cross-domain diagnostic, not just within RPMs. The biology and chemistry domains have no counterpart in I-RAVEN-X.

The RPT extension (Section 4.1, lines 210-216) pushing RC beyond the 2D RPM theoretical limit of 4 is a nice design choice that enables testing at higher RC values.

Weaknesses:

The distractor generation strategy for REL-A is underdocumented. Algorithm 1 (Appendix A.1, line 27) uses RandomDistractor() without specifying the distribution. With 8-choice format (12.5% chance), if distractors are systematically far from the correct answer in numerical value, models could use elimination rather than relational reasoning. I don't think this is necessarily a fatal flaw—the RC gradient would likely still hold even with some elimination effect—but the paper should report the average numerical distance between the correct answer and nearest distractor to rule this out. This is a quick fix.

Only three closed-source API models are evaluated. This limits reproducibility (API versions change) and prevents testing whether RC bottlenecks differ across model families, scales, or training approaches. Even one open-source model (e.g., Llama-3-70B) on a REL-A subset would strengthen the "model-agnostic" claim substantially.

Figures 5-10 lack error bars or confidence intervals. Some subsets are small—REL-C2 at N_molecules=50 has only 47 questions (Table A.2). Reporting precise completion rate differences (e.g., "38.1% vs 26.0%") on 47 samples without any uncertainty measure is not great practice.

The RC comparability across domains is assumed but not argued. Algebra RC ranges from 1-6, biology N_ht from 2-25, chemistry RC jumps to ~29 for C3. The paper treats these as instances of the same concept, but whether simultaneously tracking 4 RPM constraints is cognitively equivalent to tracking 25 homoplastic taxa is an open question. The core finding—"within each domain, higher RC means worse performance"—doesn't require cross-domain comparability, so this is more of a framing issue than a substantive one.

REL-C3 reports only recall (Section 5.3, line 301-302), not precision or F1. If models generate many SMILES strings to boost recall, the metric would overstate performance. Reporting precision alongside recall would clarify whether models are "guessing" or genuinely identifying missing isomers.

The paper acknowledges limitations (lines 428-434) but briefly: multiple-choice format may obscure failures, context-length constraints cause invalid responses, tasks are synthetic. Missing from this list: closed-source model limitation, distractor design, and the fact that no mitigation strategies were tested.

---

> ### Author Rebuttal · Authors · 2026-03-30
>
> We thank the reviewer for their valuable feedback. Below we address each of the comments in detail.
>
> > 1. distractor values relative to correct answers in REL-A?
>
> In REL-A1, distractors are sampled uniformly from the same domain as the correct answer. In other settings, distractors are sampled from values in the input RPM excluding the correct answer. If fewer than seven are available, we additionally sample from the range spanned by those values, again excluding the correct answer. We will clarify this in the appendix. Although alternative strategies are possible, the average distance from the correct answer to the nearest distractor remains approximately constant across RCs (Table 1).
>
> Table 1: Average absolute distance between the correct answer and the closest distractor in REL-A3 for different rule types.
> | Rule Type (RC) | 3x3  | 9x9  | 15x15 | 30x30 |
> |----------------|------|------|-------|-------|
> | REL-A3 - Permutation (n) | 24.33 | 29.96 | 31.53 | 32.26 |
>
> > 2. open-source model on a REL-A subset
>
> We agree with the reviewer that investigating smaller open-weight models would be a valuable extension to the current evaluation. Below, we present additional results with Qwen3-4B and Llama3.1-70B on a subset of the REL-A tasks (Table 2). Both models support our main claim that relational complexity tracks performance better than input size, though Llama is stronger than Qwen. We will test both models on all tasks for the final manuscript.
>
> Table 2: Performance on open source models for 500 questions from REL-A2 and 500 questions from REL-A3.
> | Model | Rule Type (RC) | 3x3 | 9x9 | 15x15 | 30x30 |
> |-------|----------------|-----|-----|-------|-------|
> | Llama 3.1-70b | Progression (2) | 0.912 | 0.864 | 0.856 | 0.816 |
> | Llama 3.1-70b | Permutation (n) | 0.648 | 0.552 | 0.296 | 0.144 |
> | Qwen3-4B | Progression (2) | 0.616 | 0.568 | 0.528 | 0.504 |
> | Qwen3-4B | Permutation (n) | 0.552 | 0.216 | 0.184 | 0.104 |
>
> > 3. For REL-C3, what are the precision values
>
> To address the concern that recall may reward over-generation, we also computed precision and F1 for REL-C3 (Table 3). Performance is not explained by indiscriminate guessing: mean precision is 0.344 and mean F1 is 0.252 across models and questions, and F1 declines with recall as RC increases with molecule count. This supports our interpretation that REL-C3 captures genuine difficulty in identifying missing isomers rather than a metric artifact.
>
> | Task | Model | count | n_mol_bin | Recall | Precision | F1 |
> |------|-------|-------|-----------|--------|-----------|----|
> | C3 | Claude Opus 4.5 | 300 | <20 | 0.341 (0.013) | 0.569 (0.016) | 0.386 (0.011) |
> | C3 | Claude Opus 4.5 | 400 | 20-40 | 0.313 (0.011) | 0.390 (0.013) | 0.299 (0.008) |
> | C3 | Claude Opus 4.5 | 300 | >=40 | 0.253 (0.009) | 0.398 (0.015) | 0.289 (0.009) |
> | C3 | Gemini 3 Pro | 300 | <20 | 0.417 (0.014) | 0.472 (0.015) | 0.403 (0.012) |
> | C3 | Gemini 3 Pro | 400 | 20-40 | 0.362 (0.012) | 0.275 (0.011) | 0.260 (0.009) |
> | C3 | Gemini 3 Pro | 300 | >=40 | 0.259 (0.010) | 0.256 (0.012) | 0.234 (0.010) |
> | C3 | GPT 5.2 | 300 | <20 | 0.142 (0.011) | 0.410 (0.019) | 0.173 (0.010) |
> | C3 | GPT 5.2 | 400 | 20-40 | 0.140 (0.008) | 0.198 (0.010) | 0.130 (0.006) |
> | C3 | GPT 5.2 | 300 | >=40 | 0.102 (0.006) | 0.186 (0.011) | 0.124 (0.007) |
>
> > 4. tested other forms of test-time compute
>
> We also performed an exploratory analysis of test-time compute on REL-B1. We sampled 15 questions at each of four RC levels given by the number of homoplastic taxa ($N_{ht}\in{4,10,20,25}$) and ran GPT-5.2 five times per question. We compared best-of-five and majority vote. Best-of-five improved accuracy only at low RC (46.7% to 60.0% at $N_{ht}=4$) and had little effect otherwise (no gain at 10 or 20; 0% to 6.7% at 25). Majority vote generally reduced performance. Although limited in scale, these results suggest that simple self-consistency-style test-time compute does not materially mitigate degradation at high RC.
>
> > error bars or confidence intervals
>
> We agree and will update the figures accordingly.
>
> > RC comparability across domains is assumed but not argued
>
> Performance differences at fixed RC reflect operand complexity (OC). We will clarify that RC is a shared structural notion, whereas OC is domain- and representation-dependent. Our core empirical claim is therefore primarily within-domain: increasing RC degrades performance, while OC explains additional within-domain variation.
>
> > Limitations
>
> We agree that the discussion of limitations should be expanded. We also address some of the limitations raised by including our experiments on evaluating REL-A on open-source models, clarifications on distractor design in REL-A, additional forms of test-time compute mitigation strategy for REL-B, and REL-C3 precision and F1 scores in the final manuscript. We agree that we have not provided a comprehensive overview of mitigation strategies, and will include this as a limitation of our current study.

---

> > ### Author Rebuttal · Reviewer_e2ke · 2026-04-01
> >
> > Thank you for the detailed rebuttal. The authors addressed my main questions well, especially by adding distractor statistics for REL-A, open-source model results, precision/F1 for REL-C3, and an exploratory analysis of additional test-time compute strategies. These additions strengthen the empirical support for the paper’s main claims.
> >
> > Some broader questions remain natural future work, but I believe the concerns raised in my review have been adequately addressed. I therefore consider them fully resolved and keep my overall assessment unchanged.

---

### Official Review · Reviewer_ifmF · 2026-03-07

**Soundness:** 3
**Presentation:** 4
**Significance:** 2
**Originality:** 3
**Overall Recommendation:** 4
**Confidence:** 4

**Summary:**

This paper introduces Relational Complexity (RC), borrowed from cognitive science, as a difficulty axis for evaluating LLM reasoning. RC is defined as the minimum number of independent entities that must be simultaneously bound to apply a relation. The authors build REL, a generative benchmark with tasks in algebra (Raven's Progressive Matrices/Tensors), biology (homoplasy detection in phylogenetic trees), and chemistry (isomer classification, maximum common substructure, isomer completion). They evaluate frontier models like Claude Opus 4.5, Gemini 3 Pro Preview, and GPT-5.2 and observe monotonic performance degradation as RC increases, even when entity count and input size are controlled. The trends do not change even with increased test-time compute and in-context learning.

**Compliance With Llm Reviewing Policy:**

Affirmed.

**Final Justification:**

Considering the discussion and the rebuttal, I increase my score.

**Key Questions For Authors:**

Can structured reasoning reduce effective RC? (Prompting explicitly to decompose tasks with few-shot examples) If a model decomposes a high-RC task into sequential lower-RC subtasks (e.g., checking homoplasy for pairs of taxa sequentially rather than all at once), does performance recover? If yes, the limitation is not about a fundamental capability gap, which significantly changes the paper's message. If not, that would be a much stronger result.
How correlated are your regression variables in REL-B? Please report variance inflation factors or pairwise correlations between N_ht, motif ratio, sequence length, pairwise distance, and prompt length. If N_ht is strongly correlated with prompt length or pairwise distance, the variance decomposition is less interpretable.
Why not test models with tool calls? For REL-A, a model with a Python interpreter could trivially solve all tasks by writing code. Does this mean RC is measuring in-context reasoning limitations specifically, rather than fundamental capability? If so, the paper should be explicit about this scope limitation.
Can you formalize OC? The paper introduces OC informally, but it clearly matters (C1 vs C2 gap). Without a formal definition, you have two difficulty axes, but only one is principled. Could you provide a definition of OC analogous to the RC definition, and show the two are independent?
What happens with finetuning on high-RC instances? Even a small-scale experiment (e.g., finetuning an open model on REL-A tasks with RC=3-4 and testing on RC=6+) would tell us whether this is a training distribution issue or something deeper.
Can the authors test with Answer Matching [1]  instead of the MCQ format? since it's a very well-known pitfall with a lot of confounders for evaluating LLMs. Since some of the high-RC results are close to the random-guess rate in MCQs, we cannot be sure whether they reflect actual degradation or are confounded.
[1] Chandak, Nikhil, et al. "Answer matching outperforms multiple choice for language model evaluation." arXiv preprint arXiv:2507.02856 (2025)

**Limitations:**

Yes, the authors discuss limitations honestly in Section 6 (synthetic tasks, multiple-choice evaluation, context length constraints). They could additionally discuss: (a) that tool call models might trivially solve several of these tasks, (b) that OC remains informalized, and (c) that the benchmark may be measuring in-context working memory rather than “relational reasoning” in the broader cognitive sense.

**Strengths And Weaknesses:**

Strengths:
The formalization of RC as a difficulty axis for measuring LLM reasoning is clean and well-motivated. The paper's presentation is great, and the figures (especially Figure 1) and tables clearly communicate the message.
The benchmark does not feel like a forced amalgamation of multiple domains but is well-executed and coherent. Tasks are diverse from various domains like Algebra, Biology, and Chemistry, and they do not trivially involve the same types of relational reasoning.
There are sound empirical contributions. The variance decomposition in REL-B (Fig. 7) is a very convincing analysis. Showing that RC explains 24-44% of the variance, while the next-best factor explains 6-17% seems convincing to me. The observation that performance on REL-A5/A6 improves with input size (because more data helps when RC is fixed), while REL-A3/A4 degrades with input size (because RC scales with size), provides a clear disambiguation.

Weaknesses:
Albeit, the central claim that simultaneously satisfying more constraints is harder is expected. Transformers have finite working memory in their hidden states, and higher-arity constraint satisfaction requires more simultaneous binding. The paper carefully measures this, but doesn't explain why it happens in terms of architecture or representations, nor does it propose any fixes.
The paper introduces Operand Complexity as a secondary axis to explain why C1 and C2 (both RC=2) have very different performance levels. If OC can cause performance to drop from 65.7% to 38.1% at the same RC, then RC alone is probably insufficient as a difficulty measure. The paper's core claim that “performance tracks RC more reliably than traditional proxies” is weakened. The OC axis appears underdeveloped (though acknowledged) and requires further explanation.
There isn't enough clarity about scaling the test time compute experiment. I’m assuming that since all of these are frontier reasoning models, they could very well use tool calls to offload some computation. My bet would be that with access to the right tools, like a Python interpreter for algebra tasks, they could perform much better. Is this actually the case? Are the extra tokens allocated being used for the tool calls, or do they simply contribute to the thinking tokens? This matters because the obvious hypothesis is that RC measures the width of simultaneous binding, and sequential decomposition (if possible) should reduce effective RC.
The paper bundles a neat failure analysis but provides no insight into what would fix it. Does finetuning on high-RC tasks help? Does explicit working memory (retrieval-augmented approaches, scratchpad) help? Is this a fundamental expressivity limitation of transformers, or a training data distribution issue? How does it relate to Transformer Complexity [1]? Do high-RC tasks fall into a complexity class that CoT provably helps with, or whether the simultaneous binding requirement mean serial decomposition doesn't reduce the effective complexity?
In the biology task, RC = N_ht (number of homoplastic taxa). But increasing N_ht also increases the combinatorial space the model must search (more taxa to identify = more outputs), the tree-traversal complexity, and the number of comparisons needed. The variance decomposition partially addresses this, but the regression model treats these as independent variables when they are actually correlated with N_ht. The paper should discuss collinearity diagnostics.
[1] Merrill, William, and Ashish Sabharwal. "The expressive power of transformers with chain of thought." arXiv preprint arXiv:2310.07923 (2023)

---

> ### Author Rebuttal · Authors · 2026-03-30
>
> We thank the reviewer for their detailed and constructive feedback and answer their main questions below.
>
> > Can structured reasoning reduce effective RC? [,,,] If not, that would be a much stronger result.
>
> We do not think “structured decomposition” is a general way around high RC as we define it. RC is the minimal number of independent sources that must be bound simultaneously in a reasoning step, so if a task can be reformulated into a correctness-preserving sequence of lower-RC subtasks, then its RC was overstated to begin with, rather than “reduced” by prompting.
>
> To test whether the RC bottleneck reflects a failure to execute an appropriate reasoning procedure, we introduced a structured prompting strategy in REL-B that scaffolds the solution into explicit stages (motif scanning → consistency filtering → tree-distance check → final decision). This does not change the task’s RC, but may help the model organize the computation. We evaluate this on REL-B instances with increasing RC (4, 10, 20, 25). Structured prompting improves performance at low-to-moderate RC (e.g., 45%→65% at RC=4, 15%→25% at RC=10), but gains diminish at higher RC (5%→10% at RC=20, no improvement at RC=25). These results suggest that procedural scaffolding helps when the simultaneous-binding demand is modest, but does not remove the degradation at high RC. Thus, the bottleneck is not simply the absence of a reasoning procedure, but the difficulty of maintaining multiple interacting variables simultaneously.
>
> > The variance decomposition partially addresses this, but the regression model treats these as independent variables when they are actually correlated with N_ht. The paper should discuss collinearity diagnostics.
>
> We also checked collinearity in REL-B using GVIF. Results were low for the main factors (homoplastic taxa 1.17, distance bin 1.18, motif ratio bin 1.30). Sequence length and prompt bin were higher (7.23, 5.37), as expected because longer sequences yield longer prompts. Thus the variance decomposition is not driven by collinearity between RC and the other main factors.
>
> > Why not test models with tool calls? [...] Does this mean RC is measuring in-context reasoning limitations specifically, rather than fundamental capability? If so, the paper should be explicit about this scope limitation.
>
> We agree that tool access is an important distinction: our main results measure unassisted in-context relational reasoning, not the full capability of a tool-augmented system. To test whether the bottleneck in REL-C3 is mainly due to low-level molecular parsing or chemistry operations, we ran REL-C3 with RDKit access on all questions. Performance remained poor: precision increased slightly, but recall remained low (mean 0.094) and F1 remained low (0.137). This suggests that the degradation is not eliminated by externalizing molecular parsing and chemistry operations.
>
> | Task | Count | Model | n_mol | Recall | Precision | F1 |
> |------|-------|-------|-------|-|-|-|
> | C3 | 300 | gpt-5.4 & tools | <20 | 0.109 (0.009) | 0.638 (0.021) | 0.160 (0.009) |
> | C3 | 400 | gpt-5.4 & tools | 20-40 | 0.094 (0.006) | 0.428 (0.018) | 0.130 (0.007) |
> | C3 | 300 | gpt-5.4 & tools | >=40 | 0.079 (0.005) | 0.470 (0.020) | 0.125 (0.006) |
>
> Together with the structured-prompting results, this supports a consistent interpretation: better procedural scaffolding and external tools can help at the margins, but they do not remove the core high-RC failure. The difficulty is therefore not simply that the model lacks an explicit procedure or cannot manipulate the underlying representation, but that it struggles to coordinate multiple interacting variables when the task imposes an irreducible simultaneous-binding requirement.
>
> > The OC axis appears underdeveloped and requires further explanation.
>
> For a step with RC=k, OC measures how hard it is to identify or process the k operands individually, whereas RC measures how many must be jointly integrated. Thus, tasks can share RC but differ in OC: REL-C1 and C2 are both binary, but C2 has higher OC because it requires finding the largest common substructure rather than matching formulae, consistent with the drop from 65.7% to 38.1%.
>
> > What happens with finetuning on high-RC instances?
>
> We agree that this is a very interesting question but consider this to be clearly out of scope for the present paper. For results with open weight models that one could in theory finetune on e.g. the REL-A tasks, please see the other reviewers.
>
> > Can the authors test with Answer Matching instead of the MCQ format?
>
> We additionally tested direct-answer generation on REL-A2/A3 (no answer options) with GPT-5.2. Accuracy is slightly lower than in MCQ, but the RC trend is unchanged. We will include these non-MCQ results in the revision.
>
> | Rule Type (RC) | 3x3 | 9x9 | 15x15 | 30x30 |
> |-|-:|-:|-:|-:|
> | Progression (2) | 1.000 | 1.000 | 0.928 | 0.872 |
> | Permutation (n) | 0.960 | 0.856 | 0.664 | 0.504 |

---

### Official Review · Reviewer_WwBX · 2026-03-07

**Soundness:** 4
**Presentation:** 3
**Significance:** 3
**Originality:** 3
**Overall Recommendation:** 5
**Confidence:** 3

**Summary:**

The paper studies relational reasoning. In particular, the focus is on problems where constraints involving multiple objects have to be exploited to infer the answer. The central hypothesis in the paper is that the complexity of such problems, for LLMs, depends on the number of objects that simultaneously need to be considered, which the authors call "relational complexity" (following work from cognitive science). Based on this idea, the paper introduces a benchmark involving three domains: a generalisation of raven progression matrices (which is referred to as "relational reasoning in algebra" in the paper), a biologically inspired dataset which requires identifying patterns in DNA sequences, and a chemically inspired dataset which requires identifying patterns in molecules. Experiments on three frontier LLMs support the main hypothesis, i.e. problems with higher relational complexity seem to cause more problems for LLMs.

**Compliance With Llm Reviewing Policy:**

Affirmed.

**Final Justification:**

This is nice work which, in my view, deserves to be accepted. The minor comments I mentioned in my original review have been addressed well in the rebuttal.

**Key Questions For Authors:**

The evaluation focuses on three frontier models. It would be interesting to see an extension of the evaluation to smaller open-weight models. Perhaps the hardest problems would be too difficult, but some of the Raven Progression Tensors could probably solved using open-weight models as well.

**Limitations:**

yes

**Strengths And Weaknesses:**

The paper makes important points about the difficulties in properly evaluating the reasoning capabilities of LLMs (which are not entirely new but nonetheless deserve more attention and are well-motivated in the paper). The proposed benchmark is interesting and well-considered, with systematic variation in the relevant dimensions of difficulty. The fact that three different domains are included is also commendable.

The Raven Progression Tensors offer a clear and easily-extensible framework for evaluating LLMs, complementing existing benchmarks in a useful way.

The datasets that were introduced for the other two domains also seem interesting, but the explanations of these reasoning problems can be improved: due to my limited background in biology and chemistry, I found the explanations rather hard to follow. I appreciate that there is some "additional background" in the appendix, but even this was not sufficient. I think it should be possible to better explain these reasoning tasks, such that they are clearer to a broader audience.

I think the related work section can be improved in terms of better explaining what existing work aims to evaluate. Many previous benchmark are indeed focused on reasoning about knowledge-graph (KG) like structures. This KG-based setting is indeed simpler in many respects, but the difficulty often comes from the need to do multi-hop reasoning. The relations of interest are just binary, but the total number of objects that simultaneously need to be considered is thus much higher (e.g. 10 hop inference with binary relations typically involves 11 entities in total). Some of the difficulties that arise in that setting relate to systematic and compositional generalization. This is all very different from this paper, but I think the related work section could acknowledge these orthogonal aspects of difficulty a bit better (and perhaps comment on the similarities and differences between multi-hop reasoning and high "relational complexity").

Minor points:

* In RPM3 in Fig. 3, I don't see why we have RC=4. To infer the answer, we only need to look at one other object from the same row and one other object from the same column. Doesn't that mean RC=3?

* In the explanation of A5: "Each entry is the sum of the same 4 predecessors along the x-, y-, and z-axis" was not fully clear (although I understood the main point). There are 3 dimensions but we need 4 precedessors. So do we pick one dimension where there are 2 predecessors and two dimensions where there is 1 predecessor? Or could we also e.g. have 4 predecessors in one dimensions and none in the others?

---

> ### Author Rebuttal · Authors · 2026-03-30
>
> We thank the reviewer for their very positive and constructive feedback.
>
> > The datasets that were introduced for the other two domains also seem interesting, but the explanations of these reasoning problems can be improved: due to my limited background in biology and chemistry, I found the explanations rather hard to follow. I appreciate that there is some "additional background" in the appendix, but even this was not sufficient. I think it should be possible to better explain these reasoning tasks, such that they are clearer to a broader audience.
>
> We will add the following additional content to the appendix:
>
> - REL-B1 A classic example of homoplasy is the independent evolution of gliding in distantly related mammals, such as flying squirrels and sugar gliders. These animals share a similar adaptation because similar environmental pressures favored it, not because they inherited that trait from a recent common ancestor. The same idea can occur at the sequence level: similar mutations may arise independently in different lineages because they are favored by similar selective pressures. When the same or similar sequence changes appear in different lineages for reasons other than direct inheritance from a shared ancestor, this is homoplasy.
>
> - REL-C1 (constitutional isomer set classification). Here the model is asked a simple question: do these molecules all have the same molecular formula? “Same formula” refers to molecules with the same number of each type of atom overall. “Different connectivity” means those same atoms can be wired together in different ways. So this task asks whether the moelcules are different rearrangements of the same atoms.
>
> - REL-C2 (largest connected common chemical motif). Here the model must find the biggest connected piece that appears in every molecule in the set. A reader can think of this as identifying the common backbone shared by all examples. This is harder than C1 because it is no longer enough to count atoms; the model must compare how atoms are connected and isolate the largest shared pattern.
>
> - REL-C3 (missing isomer completion). Here the model is shown only part of an isomer family and must determine which valid family members are missing. This is harder because the model must reason jointly over the shared formula, the possible space of valid structures for that formula, and the subset already shown.
>
> > I think the related work section can be improved...
>
> We agree that the related-work discussion should better separate RC from other reasoning demands studied in KG and multi-hop benchmarks. Our claim is not that those benchmarks are easier or less important, but that they probe a largely orthogonal axis. As noted by the reviewer, multi-hop settings often stress sequential composition, systematic generalization, and reasoning over longer chains of binary relations, whereas RC measures the minimal number of independently varying operands/entities that must be bound simultaneously in a single reasoning step. We will revise the related-work section to make this distinction explicit and to discuss RC and multi-hop reasoning as complementary, not competing, notions of difficulty.
>
> > In RPM3 in Fig. 3, I don't see why we have RC=4.
>
> For RPM3, the intended RC is 4 under our definition because the blank must satisfy the row and column constraints jointly, which requires integrating the two known cells in the row with the two known cells in the column. We will revise the explanation to state this more carefully.
>
> > In the explanation of A5 [...] could we also e.g. have 4 predecessors in one dimensions and none in the others?
>
> The idea is to look at neighbors whose x-, y-, or z- coordinate is the same as that of the missing entry or lower by one, so we cannot have dependence on the 4 previous values along the x-axis, for example. We will make this explicit in the final manuscript.
>
> > It would be interesting to see an extension of the evaluation to smaller open-weight models.
>
> We agree with the reviewer that investigating smaller open-weight models would be a valuable extension to the current evaluation. Below, we present additional results with Qwen3-4B and Llama3.1-70B (explicitly requested by reviewer e2ke) on a subset of the REL-A tasks. The numbers from both models support the main message of the original submission, i.e. relational complexity tracks performance better than other metrics such as input size, although Llama is a clearly stronger model than Qwen here. We will test both models on all tasks for the final manuscript.
>
> - Llama3.1-70B:
>
> | Rule Type (RC)  |   3x3 |   9x9 | 15x15 | 30x30 |
> | --------------- | ----: | ----: | ----: | ----: |
> | Progression (2) | 0.912 | 0.864 | 0.856 | 0.816 |
> | Permutation (n) | 0.648 | 0.552 | 0.296 | 0.144 |
>
> - Qwen3-4B:
>
> | Rule Type (RC)  |   3x3 |   9x9 | 15x15 | 30x30 |
> | --------------- | ----: | ----: | ----: | ----: |
> | Progression (2) | 0.616 | 0.568 | 0.528 | 0.504 |
> | Permutation (n) | 0.552 | 0.216 | 0.184 | 0.104 |

---

> > ### Author Rebuttal · Reviewer_WwBX · 2026-04-01
> >
> > This is nice work which deserves to be accepted. The minor comments I mentioned in my original review has been addressed well in the rebuttal.

---

### Decision · Program_Chairs · 2026-04-30

**Decision:**

Accept (regular)

**Comment:**

Reviewers reached a clearly positive consensus after rebuttals and discussion. The main strengths are a clean and useful formulation of relational complexity as a diagnostic axis for LLM reasoning, a well-designed generative benchmark spanning three distinct domains, and consistent within-domain evidence that performance degrades as RC increases while simpler proxies explain less. The rebuttal addressed the main empirical concerns by adding open-weight model results, non-MCQ checks, distractor statistics, precision/F1 for REL-C3, structured prompting, extra test-time compute analyses, tool-assisted results, and a clearer discussion of scope and operand complexity. The remaining limitations are mainly about breadth and framing rather than validity: broader open-model coverage would help; uncertainty reporting should be improved; and the paper should state more explicitly that the results concern unassisted in-context reasoning and that absolute RC comparability across domains is less important than within-domain trends.